# Assuming Locally Equal Calibration Errors for Non-Parametric Multiclass Calibration

**Kaspar Valk**                                                                                     *valk.kaspar@gmail.com*

**Meelis Kull**                                                                                       *meelis.kull@ut.ee*
*University of Tartu*

**Reviewed on OpenReview:** *https://openreview.net/forum?id=na5sHG69rI*

## Abstract

A probabilistic classifier is considered calibrated if it outputs probabilities equal to the expected class distribution given the classifier's output. Calibration is essential in safety-critical tasks where small deviations between the predicted probabilities and the actually observed class proportions can incur high costs. A common approach to improve the calibration of a classifier is to use a hold-out dataset and a post-hoc calibration method to learn a correcting transformation for the classifier's output. This work explores the field of post-hoc calibration methods for multi-class classifiers and formulates two assumptions about the probability simplex which have been used by many existing non-parametric calibration methods, but despite this, have never been explicitly stated: assuming locally equal label distributions or assuming locally equal calibration errors. Based on the latter assumption, an intuitive non-parametric post-hoc calibration method is proposed, which is shown to offer improvements to the state-of-the-art according to the expected calibration error metric on CIFAR-10 and CIFAR-100 datasets.

## 1 Introduction

Probabilistic classifiers take some data as input and produce probability distributions over classes as output. For example, a classifier could be tasked to take as input an X-ray image of a person's chest and produce as output a vector of three probabilities for whether the image depicts a *healthy lung, lung cancer* or *some other lung disease.* A classifier is considered to be calibrated if its predicted probabilities are in correspondence with the true class distribution. It is possible that a probabilistic classifier is not well-calibrated and produces distorted probabilities. For example, predicting an X-ray image to show a *healthy lung* with a probability of 0.9 is calibrated if, among a large sample of images with similar predictions, 0.9 of them truly depict a healthy lung. If in reality only 0.7 of the images depict a healthy lung, then the prediction of 0.9 is over-confident. The problem of over-confident predictions is especially common for modern deep neural networks (Guo et al., 2017; Lakshminarayanan et al., 2017). Distorted output probabilities are also characteristic of many classical machine learning methods such as naive Bayes, decision trees (Niculescu-Mizil & Caruana, 2005; Domingos & Pazzani, 1996) or high-dimensional logistic regression (Bai et al., 2021; Clarté et al., 2022a;b).

Well-calibrated classifiers are essential in safety-critical applications (e.g. medical diagnosis, autonomous driving) where small deviations of predicted probabilities from being calibrated can cause costly mistakes (Leibig et al., 2017). For example, in a self-driving car that uses a classifier to detect if the road is clear of obstructions, over-confident predictions can lead to accidents, and under-confident predictions can prevent the vehicle from driving.

The machine learning literature has two fundamentally different approaches to achieve better-calibrated classifiers. The first approach, with a focus on neural networks, is to modify the classifier's training algorithm or use Bayesian approaches to model uncertainties. For example, Mukhoti et al. (2020) studied the use of focal loss (Lin et al., 2017) instead of log-loss for training better calibrated classifiers; Müller et al. (2019) investigated the use of label smoothing (Szegedy et al., 2016) during training for better calibration; Kumar et al. (2018) and Popordanoska et al. (2021) proposed additional terms to be added to the training time loss function that penalize miscalibration; Maddox et al. (2019) proposed Bayesian model averaging for achieving calibration in deep learning.

The second approach to achieve better-calibrated classifiers is to apply a *post-hoc calibration method* on an already trained classifier. Post-hoc calibration methods receive as input a classifier and a hold-out validation dataset and learn a transformation from the classifier's predictions to better-calibrated predictions. Many methods have been proposed for binary probabilistic classifiers, where the output has only two classes and only one degree of freedom. For example, there exists logistic calibration (Platt, 1999); isotonic calibration (Zadrozny & Elkan, 2002); histogram binning (Zadrozny & Elkan, 2001); beta calibration (Kull et al., 2017). For multi-class classifiers with more than two output classes, a common approach has been to apply binary methods in a one-versus-rest manner: a binary post-hoc calibration method is applied separately on the probabilities of each class (Zadrozny & Elkan, 2002). In recent years, several inherently multi-class post-hoc calibration methods have been proposed as well, even though some of them are applicable only for neural networks. For example, Guo et al. (2017) proposed temperature scaling, vector scaling, and matrix scaling; Kull et al. (2019) introduced Dirichlet calibration; Zhao et al. (2021) proposed a method specifically intended for decision making scenarios; Rahimi et al. (2020) suggested intra-order preserving functions; Wenger et al. (2020) proposed a non-parametric method based on a latent Gaussian process.

This work takes a look at the *post-hoc calibration method* approach for achieving better calibrated multi-class classifiers. While there already exist many strong multi-class methods, several of them are limited to symmetrical transformations for all the classes; for example, temperature scaling, Gaussian process calibration, diagonal and order-invariant subfamilies of the intra-order preserving functions are all symmetrical. As shown in the experiments section, symmetrical transformations are usually not a problem but can be severely limiting in some cases. The asymmetrical existing methods are limited in their expressivity; for example, matrix scaling and vector scaling are limited to only linear transformations in the logit space. This work proposes an intuitive and simple non-parametric post-hoc calibration method that is not limited by a symmetrical transformation or the expressivity of parametric function families. The basis of the proposed method is assuming that similar predictions on the probability simplex have similar calibration errors. The method is shown to outperform competing methods, offering improvements in expected calibration error and avoiding the failures of symmetric methods.

In Section 2, notation is introduced and an overview of background information connected to multi-class calibration is given. The concepts of calibration, calibration error, calibration error estimation, and existing post-hoc calibration methods for multi-class calibration are explained. In Section 3, the contributions of this work are described and a post-hoc calibration method is proposed. In Section 4, experiments are carried out to compare the proposed method to its competitors. The source code of the experiments is available at `https://github.com/kaspar98/lece-calibration`. This paper builds upon preliminary research conducted for the author's master's thesis (Valk, 2022).

## 2 Background and related work

The following sections introduce the concepts of calibration, calibration error, calibration error estimation and explain the existing post-hoc calibration methods for multi-class calibration.

### 2.1 Notation

This work focuses on $m$-class classification problems with feature space $\mathcal{X}$ and one-hot encoded label space $\mathcal{Y} = \{(1, 0, \ldots, 0), (0, 1, \ldots, 0), \ldots, (0, 0, \ldots, 1)\}$, where $m \geq 3$. A probabilistic multi-class classifier for such a classification problem is a function $\boldsymbol{f} : \mathcal{X} \rightarrow \Delta^m$ that takes as input features $\boldsymbol{x} \in \mathcal{X}$ and outputs a

probability vector $\boldsymbol{f}(\boldsymbol{x}) = \hat{\boldsymbol{p}} = (\hat{p}_1, \ldots, \hat{p}_m) \in \Delta^m$, where $\Delta^m = \{(q_1, \ldots, q_m) \in [0, 1]^m | \sum_{i=1}^{m} q_i = 1\}$ is the $(m-1)$-dimensional probability simplex. In addition, let $\boldsymbol{X} \in \mathcal{X}$, $\boldsymbol{Y} = (Y_1, \ldots, Y_m) \in \mathcal{Y}$ and $\boldsymbol{f}(\boldsymbol{X}) = \hat{\boldsymbol{P}} = (\hat{P}_1, \ldots, \hat{P}_m) \in \Delta^m$ be random variables, where $\boldsymbol{X}$ denotes the input features, $\boldsymbol{Y}$ denotes the label, and $\hat{\boldsymbol{P}}$ the classifier's prediction.

## 2.2 Calibration

There exist several definitions for calibration in the context of probabilistic multi-class classifiers.

**Multi-class calibration**  A classifier is considered to be multi-class-calibrated (or just calibrated) (Kull et al., 2019) if for any prediction vector $\hat{\boldsymbol{p}} \in \Delta^m$ it holds that

$$\mathbb{E}_{\boldsymbol{Y}}\left[\boldsymbol{Y}|\hat{\boldsymbol{P}} = \hat{\boldsymbol{p}}\right] = \hat{\boldsymbol{p}}.$$

**Classwise calibration**  A weaker notion of classwise calibration conditions the expectation on each class separately (Zadrozny & Elkan, 2002). A classifier is considered to be classwise calibrated if for any class $i \in \{1, 2, \ldots, m\}$ and any real value $c \in [0, 1]$ it holds that

$$\mathbb{E}_{Y_i}\left[Y_i|\hat{P}_i = c\right] = c.$$

Note that for binary classification, classwise calibration is the same as multi-class calibration (Vaicenavicius et al., 2019).

**Confidence calibration**  Another weaker notion, confidence calibration used by Guo et al. (2017) requires calibration only for the predictions of the class with the highest probability in each output. A classifier is considered to be confidence calibrated if for any real value $c \in [0, 1]$ it holds that

$$\mathbb{E}_{\boldsymbol{Y}}\left[Y_{argmax\,\hat{\boldsymbol{P}}}|max\,\hat{\boldsymbol{P}} = c\right] = c.$$

An illustrative example of the different calibration definitions is presented in Appendix A.1.

## 2.3 Calibration error

Calibration error describes the difference between the predicted probabilities of the classifier and the corresponding perfectly calibrated class probabilities. Kumar et al. (2019) defined calibration error for confidence and classwise calibration for a classifier $\boldsymbol{f}$. In a slightly more generalized form, confidence calibration error is defined as

$$CE_{conf} = \mathbb{E}_{\hat{\boldsymbol{P}}}\left[\left|max\,\hat{\boldsymbol{P}} - \mathbb{E}_{\boldsymbol{Y}}\left[Y_{argmax\,\hat{\boldsymbol{P}}}|max\,\hat{\boldsymbol{P}}\right]\right|^{\alpha}\right]^{1/\alpha},$$

and classwise calibration error is defined as

$$CE_{cw} = \frac{1}{m}\sum_{i=1}^{m}\mathbb{E}_{\hat{\boldsymbol{P}}}\left[\left|\hat{P}_i - \mathbb{E}_{\boldsymbol{Y}}\left[Y_i|\hat{P}_i\right]\right|^{\alpha}\right]^{1/\alpha}.$$

The calibration errors are parameterized by $\alpha$, where $\alpha = 1$ results in mean-absolute-error, and $\alpha = 2$ mean-squared-error.

Calibration error could not only be defined for the whole classifier $\boldsymbol{f}$ but also for just one prediction value as the difference between the right-hand side and the left-hand side of the corresponding calibration definition. For this work, calibration error for multi-class calibration for prediction vector value $\hat{\boldsymbol{p}}$ is defined as

$$CE(\hat{\boldsymbol{p}}) = \hat{\boldsymbol{p}} - \mathbb{E}_{\boldsymbol{Y}}\left[\boldsymbol{Y}|\hat{\boldsymbol{P}} = \hat{\boldsymbol{p}}\right].$$

Note that for multi-class calibration, the error defined in such a way is a vector of real values.

### 2.4 Calibration evaluation

In any real-world setting, true calibration error can not be directly found, it can only be estimated. Common metrics to evaluate calibration are *expected calibration error* (ECE) and proper scoring rules.

The ECE metric groups similar predictions into bins and uses bin averages to estimate the calibration error. For confidence calibration, ECE is calculated using $Y_{argmax\,\hat{P}}$ and $max\,\hat{P}$ and it is defined as

$$\text{confidence ECE} = \sum_{i=1}^{b} \frac{|B_i|}{n} \cdot |\overline{p}_i - \overline{y}_i|, \tag{1}$$

where $b$ is the number of bins, $|B_i|$ the number of data points in bin $B_i$, $n$ the number of data points, $\overline{p}_i$ the average prediction value in the $i$-th bin and $\overline{y}_i$ the average label value in the $i$-th bin (Naeini et al., 2015).

For classwise calibration, ECE is defined as

$$\text{classwise ECE} = \frac{1}{m} \sum_{j=1}^{m} \text{class-}j\text{-ECE},$$

where class-$j$-ECE is calculated with the same formula as in Equation (1) but with values $Y_j$ and $\hat{P}_j$ used instead of $Y_{argmax\,\hat{P}}$ and $max\,\hat{P}$ when calculating $\overline{y}_i$ and $\overline{p}_i$ (Kull et al., 2019). Commonly bins are chosen with either equal size so that they contain an equal amount of data points, or with equal width so that the probability interval from 0 to 1 is uniformly partitioned between them. Equal-sized bins are also called equal-mass bins or data dependent bins in some works.

While ECE is an important measure for calibration evaluation, it should not be the only metric to be evaluated. Very low calibration error can be achieved if the classifier makes very uninformative predictions; e.g. predicts the overall class distribution of the training dataset for any given input. Good metrics to consider in addition to ECE are proper scoring rules (Brier score or log-loss) as they are shown to decompose into calibration loss and refinement loss (DeGroot & Fienberg, 1983). While the calibration loss measures miscalibration, the refinement loss measures the extent to which instances of different classes are getting the same prediction. The key property of proper scoring rules is to have the Bayes-optimal model as the unique loss minimizer, achieving zero calibration loss and the minimal possible refinement loss, which can be non-zero due to aleatoric uncertainty (Kull & Flach, 2015).

### 2.5 Post-hoc calibration methods

Calibration of an already trained classifier can be improved by post-hoc calibration methods. Given a classifier and a hold-out validation dataset different from the original training dataset, the goal of a post-hoc calibration method is to learn a map $\hat{c} : \Delta^m \to \Delta^m$ from the uncalibrated output $\hat{p}$ of the classifier to a better-calibrated output $\hat{c}(\hat{p})$. The ideal result would be if the calibration method learns the true calibration map $c(\hat{p}) = \mathbb{E}_Y \left[ Y | \hat{P} = \hat{p} \right]$. The transformation is typically learned by optimizing a proper scoring rule (Brier score or log-loss) (Rahimi et al., 2020). A possible motivation behind this can be that unless the refinement loss is decreasing, any reduction of a proper loss is due to the reduction of the calibration loss.

**One-versus-rest methods** A common approach to multi-class calibration has been to apply binary post-hoc calibration methods in a one-versus-rest manner (Zadrozny & Elkan, 2002). For every class in a $m$ class classification task, one can define a binary one-vs-rest classification problem: the currently viewed class is the positive class, rest of the classes grouped together are the negative class. In a one-versus-rest approach to multi-class calibration, a binary classification method is trained on $m$ such one-vs-rest tasks separately. For example, some binary calibration methods that have been applied in the one-versus-rest approach are Platt scaling (Platt, 1999), isotonic regression calibration (Zadrozny & Elkan, 2002), histogram binning (Zadrozny & Elkan, 2001), and beta calibration (Kull et al., 2017). Platt scaling and beta calibration are both parametric methods fitting a specific family for calibration, isotonic regression calibration uses isotonic

regression to learn a calibrating transformation, histogram binning divides the probability space into equal-sized bins and in each bin sets the calibrated value of predictions belonging to that bin equal to the empirical class distribution value in the bin.

There are two considerable flaws to the one-versus-rest approach. First, it is not able to learn any dependencies between classes. Second, when the output of $m$ binary methods is put back together, the prediction vector will likely no longer sum to 1 and needs to be normalized. It has been shown that normalization can make the probabilities less calibrated depending on the metric used (Gupta & Ramdas, 2022). Therefore, some works propose to ignore the final step of normalization and treat one-vs-rest calibration as truly $m$ separate binary calibration problems (Patel et al., 2021; Gupta & Ramdas, 2022).

**Temperature scaling** Temperature scaling is a logit scaling method designed for neural networks introduced by Guo et al. (2017). The method is defined as $\hat{\boldsymbol{c}}(\boldsymbol{z}) = \boldsymbol{\sigma}(\boldsymbol{z}/t)$ where $\boldsymbol{z}$ is the logit vector and $t \in (0, \infty)$ is the learned temperature parameter shared across all classes. If $t > 1$, then the method makes the predictions less confident by pulling the probability distribution towards the uniform distribution; if $t < 1$, then the method makes the predictions more confident, making the largest probability in the output even larger.

**Matrix and vector scaling** Matrix and vector scaling are both logit transformation techniques proposed by Guo et al. (2017) similar to temperature scaling. The calibrated output of these techniques is obtained by $\hat{\boldsymbol{c}}(\boldsymbol{z}) = \boldsymbol{\sigma}(\mathbf{W}\boldsymbol{z} + \mathbf{b})$, where $\mathbf{W} \in \mathbb{R}^{k \times k}$ is a matrix of learned weights (restricted to the diagonal matrix for vector scaling, unrestricted for matrix scaling) and $\mathbf{b} \in \mathbb{R}^k$ is a vector of learned biases. Vector scaling is similar to temperature scaling, but instead of a single scaling parameter, a scaling parameter is learned separately for each class and an additional bias is also learned. For matrix scaling, each logit becomes a linear combination of other logits. Matrix scaling gives better results if it is trained with off-diagonal and intercept regularization (ODIR) (Kull et al., 2019): the term $\lambda(\frac{1}{m(m-1)} \sum_{i \neq j} w_{ij}^2) + \mu(\frac{1}{m} \sum_j b_j^2)$ is added to the training loss, where $\lambda$ and $\mu$ are the regularization hyperparameters.

**Dirichlet calibration** Dirichlet calibration is a method proposed by Kull et al. (2019) that is quite similar to matrix scaling, except it does not work on the logits of a neural network but rather on the actual predicted probabilities $\hat{\boldsymbol{p}}$ of a classifier. With Dirichlet calibration, the calibrated probabilities are obtained by $\hat{\boldsymbol{c}}(\hat{\boldsymbol{p}}) = \boldsymbol{\sigma}(\mathbf{W}\ln\hat{\boldsymbol{p}} + \mathbf{b})$, where ln is a vectorized natural-logarithm function. Similar to matrix scaling, Dirichlet calibration is also trained with ODIR to prevent overfitting.

**Intra order-preserving functions** Rahimi et al. (2020) proposed to use the family of intra order-preserving (IOP) functions to learn a calibration map on the logits of a neural network. An IOP function $\hat{\boldsymbol{c}} : \mathbb{R}^m \to \mathbb{R}^m$ is a vector-valued function where the order of the sorted output components is the same as the order of sorted input components, that is $argsort\,\hat{\boldsymbol{c}}(\boldsymbol{z}) = argsort\,\boldsymbol{z}$. The use of IOP functions was motivated by their property of preserving classifier accuracy, and having larger expressive power than the scaling methods proposed by Guo et al. (2017) or Dirichlet calibration. The IOP function family preserves classifier accuracy after calibration, as the largest probability in the classifier output still remains the largest probability after calibration, thanks to the order-preserving property. The IOP function family is more expressive than matrix scaling or Dirichlet calibration, as IOP functions can learn non-linear transformations while matrix scaling and Dirichlet calibration are limited to linear transformations.

In addition, Rahimi et al. (2020) showed in their experiments that in practice, it is better to use a diagonal subfamily of IOP functions as they can be expressed with fewer parameters and are therefore less prone to overfitting. An IOP function $\hat{\boldsymbol{c}}$ is a diagonal function if $\hat{\boldsymbol{c}}(\boldsymbol{z}) = (\hat{c}(z_1), \ldots, \hat{c}(z_m))$, where $\hat{c} : \mathbb{R} \to \mathbb{R}$ is a continuous and increasing function. A diagonal IOP function is symmetrical for all classes and produces output where the different class logits do not interact with each other in $\hat{\boldsymbol{c}}$. It can be noted that temperature scaling uses a subfamily of diagonal IOP functions: it uses linear diagonal IOP functions where the bias term equals 0. Rahimi et al. (2020) implemented the IOP post-hoc calibration method as a neural network with two fully connected hidden layers with order-preserving constraints. Their implementation has two hyperparameters: the number of neurons in the first and second hidden layers.

**Decision calibration** Zhao et al. (2021) proposed a non-parametric calibration method for the context of decision making settings. The method works iteratively by partitioning the probability simplex and applying an adjustment on each partition. The partition and adjustment are both determined by the average difference between the label and prediction values.

**Gaussian process calibration** Wenger et al. (2020) proposed a natively multi-class non-parametric calibration method that uses a latent Gaussian process to learn the calibrating transformation. The method applies the same transformation for all the classes.

## 3 Contributions

The contributions of this work to the field of multi-class calibration can be split into two:

1. The work formulates two assumptions about the true calibration map that have been previously used but not clearly stated in the calibration literature.

2. By explicitly acknowledging the assumptions, we propose an intuitive and simple non-parametric post-hoc calibration method.

### 3.1 Proposed assumptions

Before introducing the proposed assumptions, a small introduction is needed. According to the definition of multi-class calibration given in Section 2.2, a prediction vector $\hat{\boldsymbol{p}}$ is considered calibrated if $\hat{\boldsymbol{p}} = \mathbb{E}_{\boldsymbol{Y}}\left[\boldsymbol{Y}|\hat{\boldsymbol{P}} = \hat{\boldsymbol{p}}\right]$. Therefore, a calibrating transformation of a prediction $\hat{\boldsymbol{p}}$ could be found if we had a good estimate of its true conditional class distribution $\mathbb{E}_{\boldsymbol{Y}}\left[\boldsymbol{Y}|\hat{\boldsymbol{P}} = \hat{\boldsymbol{p}}\right]$ — we could simply set the prediction value $\hat{\boldsymbol{p}}$ equal to the estimate. Similarly, if we were to approach calibration from the definition of calibration error $CE(\hat{\boldsymbol{p}}) = \hat{\boldsymbol{p}} - \mathbb{E}_{\boldsymbol{Y}}\left[\boldsymbol{Y}|\hat{\boldsymbol{P}} = \hat{\boldsymbol{p}}\right]$ given in Section 2.3, it would suffice for calibration if we had a good calibration error estimate — we could subtract the estimate from the prediction and our output would be calibrated.

One obvious weak estimator for the true class distribution could be the (single) label value $\boldsymbol{Y}$ corresponding to $\hat{\boldsymbol{p}}$. The estimator would clearly be unbiased as for each $\hat{\boldsymbol{p}}$, the average value of $\boldsymbol{Y}$ is equal to $\mathbb{E}_{\boldsymbol{Y}}[\boldsymbol{Y}|\hat{\boldsymbol{P}} = \hat{\boldsymbol{p}}]$, and hence, $\mathbb{E}_{\boldsymbol{Y}}[\boldsymbol{Y}|\hat{\boldsymbol{P}} = \hat{\boldsymbol{p}}] - \mathbb{E}_{\boldsymbol{Y}}[\boldsymbol{Y}|\hat{\boldsymbol{P}} = \hat{\boldsymbol{p}}] = 0$. However, this estimator $\boldsymbol{Y}$ would have very high variance as it is based on a sample with just one element. Likewise, an unbiased high variance estimator $\widehat{CE}$ could be constructed for the calibration error as the difference between the prediction and its label $\widehat{CE}(\hat{\boldsymbol{p}}) = \hat{\boldsymbol{p}} - \boldsymbol{Y}$. Unfortunately, both of these simple estimators have too high variance to be used for calibration. However, if we made some assumptions about the calibration map of our classifier, we could construct good estimators that make use of these weaker estimators.

---

**Assumption of locally equal calibration errors (LECE)** We propose to assume that the calibration error is approximately equal in a close neighborhood on the probability simplex. Formally, for some fixed $\epsilon, \delta > 0$ and some neighborhood function $d : \Delta^m \times \Delta^m \to \mathbb{R}$, we assume that

$$d(\hat{\boldsymbol{p}}, \hat{\boldsymbol{p}}') \le \delta \quad \implies \quad \left\|CE(\hat{\boldsymbol{p}}) - CE(\hat{\boldsymbol{p}}')\right\|^2 \le \epsilon$$

where $\|\cdot\|^2$ denotes the squared Euclidean norm.
*Note that the term 'locally' is often used to refer to neighborhoods in the original feature space, whereas we consider neighborhoods in the simplex.*

---

Given a validation dataset, the LECE assumption allows us to construct a considerably better estimator $\widehat{CE}_{neigh}$ for the calibration error of prediction $\hat{\boldsymbol{p}}$ than the previously introduced weak estimator $\widehat{CE}(\hat{\boldsymbol{p}}) = \hat{\boldsymbol{p}} - \boldsymbol{Y}$. First, we need to find the close neighborhood of $\hat{\boldsymbol{p}}$, meaning the validation set predictions $\hat{\boldsymbol{p}}_1, \ldots, \hat{\boldsymbol{p}}_k$ with labels $\boldsymbol{Y}_1, \ldots, \boldsymbol{Y}_k$ for which $d(\hat{\boldsymbol{p}}, \hat{\boldsymbol{p}}_i) \le \delta$ for $i$ in $1, \ldots, k$. A stronger estimator can then be constructed

if we average across the weak calibration error estimator values belonging to that close neighborhood

$$\widehat{CE}_{neigh}(\hat{\boldsymbol{p}}) = \frac{1}{k}\sum_{i=1}^{k}\widehat{CE}(\hat{\boldsymbol{p}}_i) = \frac{1}{k}\sum_{i=1}^{k}(\hat{\boldsymbol{p}}_i - \boldsymbol{Y}_i).$$

The stronger estimator has an upper bound on its squared bias as

$$Bias\left[\widehat{CE}_{neigh}(\hat{\boldsymbol{p}})\right]^2 = \left\|\mathbb{E}_{\boldsymbol{Y}_1,\dots,\boldsymbol{Y}_k}\left[\widehat{CE}_{neigh}(\hat{\boldsymbol{p}})\right] - CE(\hat{\boldsymbol{p}})\right\|^2$$

$$= \left\|\frac{1}{k}\sum_{i=1}^{k}\mathbb{E}_{\boldsymbol{Y}_i}\left[(\hat{\boldsymbol{p}}_i - \boldsymbol{Y}_i)\right] - CE(\hat{\boldsymbol{p}})\right\|^2$$

$$= \left\|\frac{1}{k}\sum_{i=1}^{k}(\hat{\boldsymbol{p}}_i - \mathbb{E}_{\boldsymbol{Y}_i}\left[\boldsymbol{Y}_i\right]) - CE(\hat{\boldsymbol{p}})\right\|^2$$

$$= \left\|\frac{1}{k}\sum_{i=1}^{k}CE(\hat{\boldsymbol{p}}_i) - CE(\hat{\boldsymbol{p}})\right\|^2 \le \frac{1}{k}\sum_{i=1}^{k}\|CE(\hat{\boldsymbol{p}}_i) - CE(\hat{\boldsymbol{p}})\|^2 \le \frac{1}{k}\sum_{i=1}^{k}\epsilon = \epsilon.$$

The variance of the estimator decreases approximately linearly as the number of neighbors increases

$$Var\left[\widehat{CE}_{neigh}(\hat{\boldsymbol{p}})\right] = Var\left[\frac{1}{k}\sum_{i=1}^{k}\widehat{CE}(\hat{\boldsymbol{p}}_i)\right]$$

$$= \frac{\sum_{i=1}^{k}Var[\widehat{CE}(\hat{\boldsymbol{p}}_i)]}{k^2} \approx \frac{Var[\widehat{CE}(\hat{\boldsymbol{p}})]}{k}.$$

Given the estimate $\widehat{CE}_{neigh}(\hat{\boldsymbol{p}})$, a calibrated prediction can finally be constructed by subtracting it from the original prediction: $\hat{\boldsymbol{c}}(\hat{\boldsymbol{p}}) = \hat{\boldsymbol{p}} - \widehat{CE}_{neigh}(\hat{\boldsymbol{p}})$.

---

**Assumption of locally equal class distributions (LECD)**  A similar derivation for constructing calibrated predictions is possible if one would instead assume that the true label distribution $\mathbb{E}_{\boldsymbol{Y}}\left[\boldsymbol{Y}|\hat{\boldsymbol{P}} = \hat{\boldsymbol{p}}\right]$ is approximately equal in a close neighborhood. Formally, for some fixed $\epsilon, \delta > 0$ and some neighborhood function $d : \Delta^m \times \Delta^m \to \mathbb{R}$, we assume that

$$d(\hat{\boldsymbol{p}}, \hat{\boldsymbol{p}}') \le \delta \quad \Longrightarrow \quad \left\|\mathbb{E}_{\boldsymbol{Y}}\left[\boldsymbol{Y}|\hat{\boldsymbol{P}} = \hat{\boldsymbol{p}}\right] - \mathbb{E}_{\boldsymbol{Y}}\left[\boldsymbol{Y}|\hat{\boldsymbol{P}} = \hat{\boldsymbol{p}}'\right]\right\|^2 \le \epsilon.$$

---

With this assumption, the method would arrive at a calibrated prediction by using the average one-hot label vector in a close neighborhood $\hat{\boldsymbol{c}}(\hat{\boldsymbol{p}}) = \frac{1}{k}\sum_{i=1}^{k}\boldsymbol{Y}_i$. Similarly to the previous LECE assumption, the estimator used with the LECD assumption would also have an upper bound of $\epsilon$ on its squared bias, and its variance would decrease approximately linearly as the number of neighbors increases.

**Visualisation of the assumptions**  To better understand the difference between the two assumptions, consider Figure 1. It depicts the calibration maps learned by two calibration methods on a synthetic calibration task. The exact details about the synthetic dataset are given in Section 4.1, but in short, 5000 prediction and label pairs were generated from a Dirichlet distribution with parameters $[0.5, 0.5, 0.5]$, and the goal of the calibration methods in Figures 1a and 1b was to learn to imitate the true calibration map depicted in Figure 1c from the generated data. Note that the true calibration map depicted in Figure 1c is something we never know in practice and here it has been manually created merely for this synthetic calibration task to allow for visual comparison between the two assumptions. Both the background colors and black arrows depict the learned transformation. The same information that is represented by the arrow is also represented by the RGB color value which is in a linear relation with the calibration error vector at that point: red for class 1, green for class 2, and blue for class 3.

(a) Histogram binning one-vs-rest
with LECD assumption (classical)

(b) Histogram binning one-vs-rest
with LECE assumption (new)

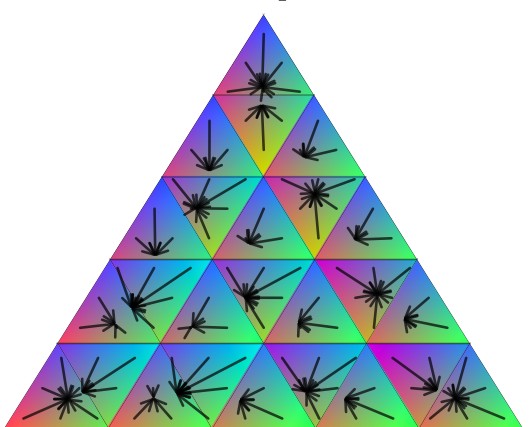 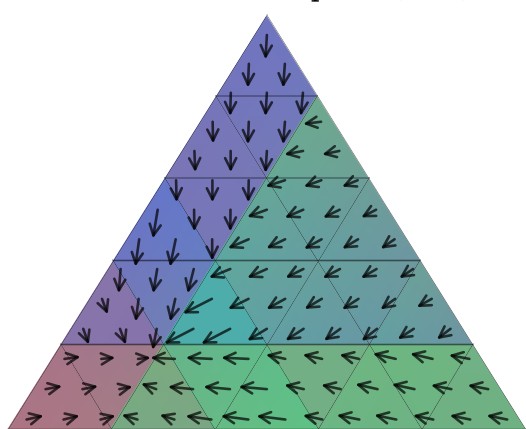

(c) True calibration map

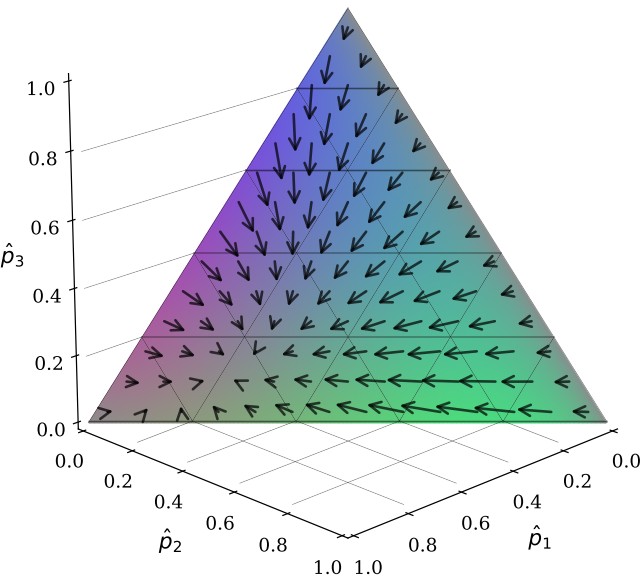

Figure 1: Illustrative example of the differences between the LECD and the LECE assumption on a synthetic calibration task. Two versions of histogram binning applied in a one-vs-rest style with 5 equal-width bins (a, b) aim to learn the true calibration map (c). The classical histogram binning uses the LECD assumption (a); the novel version uses the LECE assumption (b). Note the difference between the two methods: (a) for LECD the black calibration arrows point to the same location in one bin; (b) for LECE the arrows are the same in one bin. Histogram binning applied with the LECE assumption (b) imitates the true calibration map (c) more closely than the one applied with the LECD assumption (a). For details about the synthetic data calibration task, see Section 4.1.

Both of the methods depicted in Figure 1a and Figure 1b depict the result of a histogram binning method applied in one-vs-rest style with 5 equal-width bins. Note that the classical histogram binning uses equal-size bins, but here equal-width bins are used for a clearer visualization. The only difference between the methods is the underlying assumption used: in Figure 1a the LECD assumption is used as in classical histogram binning; the novel alternative in Figure 1b uses the LECE assumption. Note that the visualization in Figure 1a is very similar to the visualization of the higher-dimensional reliability diagrams as used by Vaicenavicius et al.

(2019), and even though they are technically the same, here Figure 1a does not depict a reliability diagram, but rather the transformation of a calibration method.

In both cases, the binning scheme defines the close neighborhoods, where we assume the corresponding assumptions to hold. With the classical LECD assumption, the end point of every calibrating arrow is the same in one bin; with the novel assumption, the calibrating arrow itself is the same in one bin. When comparing Figure 1a and Figure 1b to Figure 1c, histogram binning applied with the LECE assumption provides a closer imitation of the true calibration map than histogram binning applied with the LECD assumption. The visual intuition is also confirmed by the several numeric metrics in provided in Table 1 in Section 4.1, showing that the histogram binning based on the LECE assumption is indeed closer to the true calibration map. The LECE assumption outperformed the LECD assumption on the real experiments as well (as shown in Section 4.2.4) and is therefore preferred in this work.

**Relation to prior work** Neither of the two assumptions are completely novel as estimators based on them have been used in previous work. However, the assumptions themselves have never been explicitly stated. An estimator based on the assumption of equal calibration errors has been used in at least two algorithms that use the average difference between the label and prediction in a close neighborhood. First, the widely used ECE calculation algorithm (Naeini et al., 2015) defines the close neighborhood with bins and in each bin uses the difference between the average label and average prediction value to estimate the calibration error — this is similar to the proposed assumption as $\frac{1}{k}\sum_{i=1}^{k}\hat{\boldsymbol{p}}_i - \frac{1}{k}\sum_{i=1}^{k}\boldsymbol{y}_i = \frac{1}{k}\sum_{i=1}^{k}(\hat{\boldsymbol{p}}_i - \boldsymbol{y}_i)$. Second, the recently proposed decision calibration algorithm (Zhao et al., 2021) also uses the average difference between the predictions and labels in a close neighborhood. In the decision calibration algorithm, the close neighborhoods are defined in each iteration by the partition.

An estimator based on the assumption of equal class distributions has also been previously used. It is the basis of the histogram binning method (Zadrozny & Elkan, 2001) where the close neighborhood is defined by the binning scheme; in each bin, the average one-hot label vector is used to calibrate the predictions. The weighted average one-hot label vector is also used in recent work by Popordanoska et al. (2021) for a training time calibration method, not a post-hoc calibration method. In their work, all the neighbors of a prediction are assigned a weight with a kernel function; the weighted average of label vectors is then used to estimate the calibrated prediction.

**Defining the close neighborhood** For both assumptions, two open questions remain:

1. How many instances should the close neighborhood cover?

2. How should the close neighborhood be defined?

To answer the first question: the more neighbors taken, the less varying the estimators; however, the further away the neighbors are, the bolder the assumptions and the larger the bias from the assumption. Therefore, a sweet spot for the bias-variance tradeoff has to be found. This can be achieved if the used neighborhood scheme offers a hyperparameter defining the neighborhood size. The hyperparameter could then be optimized with cross-validation with Brier score or log-loss.

There is no simple answer to the second question. Histogram binning and the ECE algorithm define the close neighborhood with a binning scheme. However, the binning schemes are only defined for the binary case. The decision calibration algorithm (Zhao et al., 2021) defines the close neighborhoods by a linear partition that splits the probability simplex such that the calibration error estimates would be maximized. Popordanoska et al. (2021) define the neighborhood through assigning weights with a kernel function.

## 3.2 LECE calibration

One intuitive approach would be to define the close neighborhood separately for every data point: for some prediction $\hat{\boldsymbol{p}}$, the close neighborhood could be defined by the $k$ closest instances on the validation dataset. Defining the neighborhood this way results basically in a modified $k$-nearest-neighbors algorithm that we call LECE calibration. For any uncalibrated prediction $\hat{\boldsymbol{p}}$, LECE calibration would:

1. Find the $k$ closest predictions on the validation dataset $\hat{\boldsymbol{p}}_1, \ldots, \hat{\boldsymbol{p}}_k$ according to some neighborhood function $d$.

2. Estimate the calibration error of $\hat{\boldsymbol{p}}$ by finding the average difference between its neighbors' prediction and label

$$\widehat{CE}_{neigh}(\hat{\boldsymbol{p}}) = \frac{1}{k} \sum_{i=1}^{k} (\hat{\boldsymbol{p}}_i - \boldsymbol{y}_i) = \frac{1}{k} \sum_{i=1}^{k} \widehat{CE}(\hat{\boldsymbol{p}}_i).$$

3. Subtract the calibration error estimate $\widehat{CE}_{neigh}(\hat{\boldsymbol{p}})$ from the uncalibrated prediction $\hat{\boldsymbol{p}}$ to calibrate:

$$\hat{\boldsymbol{c}}(\hat{\boldsymbol{p}}) = \hat{\boldsymbol{p}} - \widehat{CE}_{neigh}(\hat{\boldsymbol{p}}).$$

Similary to the LECE calibration algorithm, a LECD calibration algorithm could be defined as well, with the only difference being in the underlying assumption used — instead of steps 2 and 3 of the algorithm, the method would instead set the calibrated prediction equal to the average label value of the neighbors $\hat{\boldsymbol{c}}(\hat{\boldsymbol{p}}) = \frac{1}{k} \sum_{i=1}^{k} \boldsymbol{y}_i$.

For the neighborhood function $d$, we considered Kullback-Leibler divergence and Euclidean distance. As shown in the real data experiments in Section 4.2.4, Kullback-Leibler divergence performed slightly better. To find the neighbors of $\hat{\boldsymbol{p}}$, Kullback-Leibler divergence is applied as $d_{\mathsf{KL}}(\hat{\boldsymbol{p}}, \cdot)$ where $d_{\mathsf{KL}}(\hat{\boldsymbol{p}}, \hat{\boldsymbol{p}}') = \sum_{i=1}^{m} \hat{p}_i \log\left(\frac{\hat{p}_i}{\hat{p}'_i}\right)$, and the term in the sum is considered equal to 0 if $\hat{p}_i = 0$ and equal to $\infty$ if $\hat{p}_i \neq 0$ and $\hat{p}'_i = 0$. The number of classes is denoted by $m$.

**Thresholding tiny probabilities**   A problem inherent to the non-parametric LECE (and LECD) calibration method is its inability to work well for tiny probabilities. This is because the method uses an estimator, which has some built in errors coming from its bias and/or variance. For class probabilities that are very near to 0, these errors of the estimator become very large proportionally to the probability. To see this, consider a true class probability $p_i$ estimated based on $k$ neighbours. In the ideal case where all the neighbours have the same true label distribution, the variance of this estimator is $\frac{p_i(1-p_i)}{k}$. Hence the estimator's relative error (standard deviation divided by the true value) is $\frac{\sqrt{p_i(1-p_i)}}{\sqrt{k}}/p_i = \frac{\sqrt{1-p_i}}{\sqrt{p_i \cdot k}}$ which becomes increasingly large when $p_i$ gets small. This could even lead to situations, where the LECE method produces output that is smaller than 0 for some classes and could no longer be interpreted as probabilities. For example, consider $\hat{p}_i = 0.01$ and suppose the average prediction of its $k$ neighbors is $\bar{p} = 0.03$ and the average label $\bar{y} = 0.01$. In that case, the calibration error estimate is $\widehat{CE}_{neigh}(\hat{p}_i) = 0.03 - 0.01 = 0.02$ and the calibrating transformation would be $\hat{c}(\hat{p}_i) = \hat{p}_i - \widehat{CE}_{neigh}(\hat{p}_i) = 0.01 - 0.02 = -0.01$, which is no longer on the probability simplex. Therefore, to overcome this problem with small probabilities, we opted to introduce one more parameter to the calibration method: a threshold value $t \in \mathbb{R}$ which sets a lower limit when to apply the method. For any class probability smaller than $t$, we do not apply the method. As the true class probability $p_i$ is unknown, then instead we apply this threshold on both the uncalibrated prediction $\hat{p}_i$ and the corresponding would-be-calibrated prediction $\hat{c}(\hat{p}_i)$.   More precisely, given the prediction vector $\hat{\boldsymbol{p}}$, and the would-be-calibrated prediction vector $\hat{\boldsymbol{c}}(\hat{\boldsymbol{p}}) = \hat{\boldsymbol{p}} - \widehat{CE}_{neigh}(\hat{\boldsymbol{p}})$, if for the $i$-th class $\hat{p}_i \leq t$ or $\hat{c}_i(\hat{\boldsymbol{p}}) \leq t$, then we set $\hat{c}_i(\hat{\boldsymbol{p}}) = \hat{p}_i$, where $\hat{\boldsymbol{c}}(\cdot) = (\hat{c}_1(\cdot), \ldots, \hat{c}_m(\cdot))$. Thresholding can cause the final output to no longer sum to 1, so to solve this, as a final step we divide the output vector by its sum. As shown by optimal threshold values chosen by hyperparameter tuning in the real experiments in Table 11, the LECE method chooses small threshold values ranging from $t = 0$ to $t = 0.02$.

**Composition with parametric methods**   The authors of the decision calibration algorithm noticed that their proposed non-parametric post-hoc calibration method works better if it is applied in composition with temperature scaling (Zhao et al., 2021). First, temperature scaling learns the calibration map on the validation dataset and then their method fine-tunes the result of temperature scaling using the temperature-scaled validation data. The benefit of composing parametric and non-parametric calibration methods was also shown by Zhang et al. (2020) who noted that isotonic regression applied in a one-versus-rest manner works better if it is applied on top of temperature scaling. A similar observation is true for the proposed

non-parametric LECE calibration method in this work as well. The experiments on real data in Section 4.2 show that the proposed method loses to existing parametric post-hoc calibration methods when applied directly, but wins when applied on top of temperature scaling. From the experiments it can be concluded that on datasets with few samples per class, LECE alone is not strong enough to compete with methods with parametric assumptions. However, LECE has its merits being less constrained than its competitors, and therefore it can offer improvements on top of parametric methods, wherever they are limited by their parametric family. Additionally, the violations of the LECE assumption are likely to diminish when the calibration errors become smaller, e.g. on top of temperature scaling.

---

**Algorithm 1:** LECE calibration method

---

**Input** : predictions on the validation set $\hat{\boldsymbol{p}}_1, \ldots, \hat{\boldsymbol{p}}_n$
          validation set labels $\boldsymbol{y}_1, \ldots, \boldsymbol{y}_n$
          prediction to calibrate $\hat{\boldsymbol{p}}$
          neighborhood size $k$
          distance function $d$
          threshold $t$
          number of classes $m$
**Output:** calibrated prediction $\hat{\boldsymbol{c}}(\hat{\boldsymbol{p}})$

**1** $D \leftarrow$ distances $d(\hat{\boldsymbol{p}}, \hat{\boldsymbol{p}}_i)$ for $i$ in $1, \ldots, n$
**2** $I \leftarrow$ indices of $k$ smallest values from $D$
**3** $\bar{\mathbf{p}} \leftarrow \frac{1}{k} \sum_{i \in I} \hat{\boldsymbol{p}}_i$                                        // average prediction
**4** $\bar{\mathbf{y}} \leftarrow \frac{1}{k} \sum_{i \in I} \boldsymbol{y}_i$                                          // average label
**5** $\widehat{CE} \leftarrow \bar{\mathbf{p}} - \bar{\mathbf{y}}$                             // calibration error estimate
**6** $\hat{\boldsymbol{c}}(\hat{\boldsymbol{p}}) \leftarrow \hat{\boldsymbol{p}} - \widehat{CE}$                      // initial calibrated prediction
**7 for** $i$ in $1, \ldots, m$ **do**
**8**     **if** $\hat{p}_i \leq t$ or $\hat{c}(\hat{\boldsymbol{p}})_i \leq t$ **then**
**9**        $\hat{c}(\hat{\boldsymbol{p}})_i \leftarrow \hat{p}_i$                       // thresholding
**10**     **end**
**11 end**
**12** $\hat{\boldsymbol{c}}(\hat{\boldsymbol{p}}) \leftarrow \hat{\boldsymbol{c}}(\hat{\boldsymbol{p}})/\sum_{i=1}^{m} \hat{c}(\hat{\boldsymbol{p}})_i$         // ensure sums to 1
**13 return** $\hat{\boldsymbol{c}}(\hat{\boldsymbol{p}})$

---

The complete pseudocode of LECE calibration with thresholding is presented in Algorithm 1. Note that if LECE calibration were to be applied in composition with temperature scaling, then the only difference in Algorithm 1 would be that the input $\hat{\boldsymbol{p}}_1, \ldots, \hat{\boldsymbol{p}}_n$ and $\hat{\boldsymbol{p}}$ would be the output of temperature scaling. A discussion on the computational and memory complexity of Algorithm 1 is given in Appendix A.2.

To summarize, the LECE calibration method is essentially a $k$-nearest neighbors algorithm using the neighbors prediction and label difference; the method involves thresholding of tiny probabilities; and it works best when composed with a parametric method.

## 4 Experiments

The experiments' section consists of two parts:

- A small experiment on synthetically generated data. The goal of this experiment is to illustrate and give visual intuition about the different post-hoc calibration methods.

- Larger experiments on two real datasets and three convolutional neural network classifiers. The goal of these experiments is to compare the proposed post-hoc calibration method with its competitors and see if it can offer improvement over the state-of-the-art.

### 4.1 Synthetic data experiment

To illustrate the differences between the assumption of locally equal calibration errors (LECE) and the assumption of locally equal class distributions (LECD), and the limitations of the existing post-hoc calibration methods, a synthetic dataset was created. A synthetic approach allows to define the true calibration function $c(\hat{p}) = \mathbb{E}_Y[Y|\hat{P} = \hat{p}]$, which is not available in any real-world dataset.

#### 4.1.1 Data generation

For the experiment, a 3-class validation dataset consisting of 5000 prediction and label pairs, and a test dataset consisting of 100000 prediction and label pairs were generated. Note that as calibration applies on the predictions and does not depend on the original feature space, we are directly generating predictions without considering the features nor the classification model at all. Classifier predictions $\hat{p}$ were sampled from a 3-class Dirichlet distribution with parameters $[0.5, 0.5, 0.5]$; the label distributions were calculated by applying the chosen true calibration function $c(\hat{p}) = \mathbb{E}_Y[Y|\hat{P} = \hat{p}]$ on the predictions; the labels $y$ were sampled from $c(\hat{p})$. The chosen true calibration function is defined as $c(\hat{p}) = (\hat{p}_1^{0.8} + \frac{\hat{p}_1 \cdot \hat{p}_2}{5}, \hat{p}_2 + \frac{\hat{p}_1 \cdot \hat{p}_3}{3}, \hat{p}_3 + \frac{\hat{p}_1 \cdot \hat{p}_2}{10})/Z$, where $Z$ is the renormalizing term to sum to 1. The chosen function is depicted in Figure 1c and repeated again in Figure 2d for convenience. Note that the results of the synthetic experiment should be considered purely illustrative as the function $c(\cdot)$ was chosen rather arbitrarily and with a different function the results could be different.

#### 4.1.2 Compared post-hoc calibration methods

On the synthetic task several post-hoc calibration methods were evaluated:

- Two different histogram binnings with 5 equal-width bins: a classical version with the LECD assumption (**H-LECD**) and a novel version with the LECE assumption (**H-LECE**). The histogram binning methods were chosen to visualise the difference between the two assumptions formalized in Section 3. The visualization between the two assumptions is provided in Figure 1.

- Temperature scaling (**TS**) to illustrate the limitations of symmetric calibration methods which can only learn transformations that act the same way for all the classes. Temperature scaling was applied to $log(\hat{p})$ as no logits were available for the task.

- Dirichlet calibration (**DIR**) to show the limited expressivity of the otherwise powerful parametric methods. Dirichlet calibration was fit without regularization as the number of parameters is low with 3 classes.

- Our proposed LECE calibration method (**LECE**) to illustrate its merits. The method was applied with $k = 500$ neighbors and threshold value $t = 0$ (thus applying thresholding only to ensure that outputs are non-negative). Note that this synthetic calibration task is meant to be purely illustrative of different calibration methods, and therefore the hyperparameters of LECE were manually chosen to show that the method can work well given good hyperparameters. The experiments on real data where hyperparameters are tuned with cross-validation show that good hyperparameter values can be found in practice as well.

#### 4.1.3 Results

The learned calibration maps of H-LECD and H-LECE are depicted in Figure 1, the maps of all other methods are shown in Figure 2. Both the background colors and black arrows in both Figure 1 and Figure 2 depict the learned transformation. The same information that is represented by the arrow is also represented by the RGB color value which is in a linear relation with the calibration error vector at that point: red for class 1, green for class 2, and blue for class 3. Table 1 contains the results of the synthetic experiment according to several different numeric metrics; it shows the mean of 100 data generation seeds with the standard deviation. For synthetic experiments, the ECE measures are replaced with the true calibration

Table 1: Results of the illustrative synthetic experiment: the mean and standard deviation over 100 data seeds is reported. For details of the applied methods, see Section 4.1.2. H-LECD (in Figure 1a) is outperformed by H-LECE (in Figure 1b). TS (in Figure 2a) is limited by symmetry and loses to DIR (in Figure 2b). DIR is limited by its parametric family and loses to LECE (in Figure 2c). The final column (true) depicts the theoretical best result which is obtained by applying the true calibration map (in Figure 2d).

|  | | ours | | | ours | |
|  | H-LECD | H-LECE | TS | DIR | LECE | true |
|---|---|---|---|---|---|---|
| confidence CE | $0.056 \pm 0.001$ | $0.019 \pm 0.002$ | $0.030 \pm 0.002$ | $0.022 \pm 0.004$ | $0.018 \pm 0.003$ | $0.000 \pm 0.000$ |
| classwise CE | $0.049 \pm 0.001$ | $0.016 \pm 0.001$ | $0.027 \pm 0.001$ | $0.018 \pm 0.002$ | $0.015 \pm 0.002$ | $0.000 \pm 0.000$ |
| Brier score | $0.448 \pm 0.001$ | $0.438 \pm 0.001$ | $0.440 \pm 0.001$ | $0.438 \pm 0.001$ | $0.438 \pm 0.001$ | $0.436 \pm 0.001$ |
| log-loss | $0.772 \pm 0.002$ | $0.749 \pm 0.010$ | $0.751 \pm 0.002$ | $0.747 \pm 0.002$ | $0.742 \pm 0.002$ | $0.738 \pm 0.002$ |
| accuracy | $0.668 \pm 0.002$ | $0.671 \pm 0.002$ | $0.669 \pm 0.001$ | $0.671 \pm 0.001$ | $0.670 \pm 0.001$ | $0.671 \pm 0.001$ |

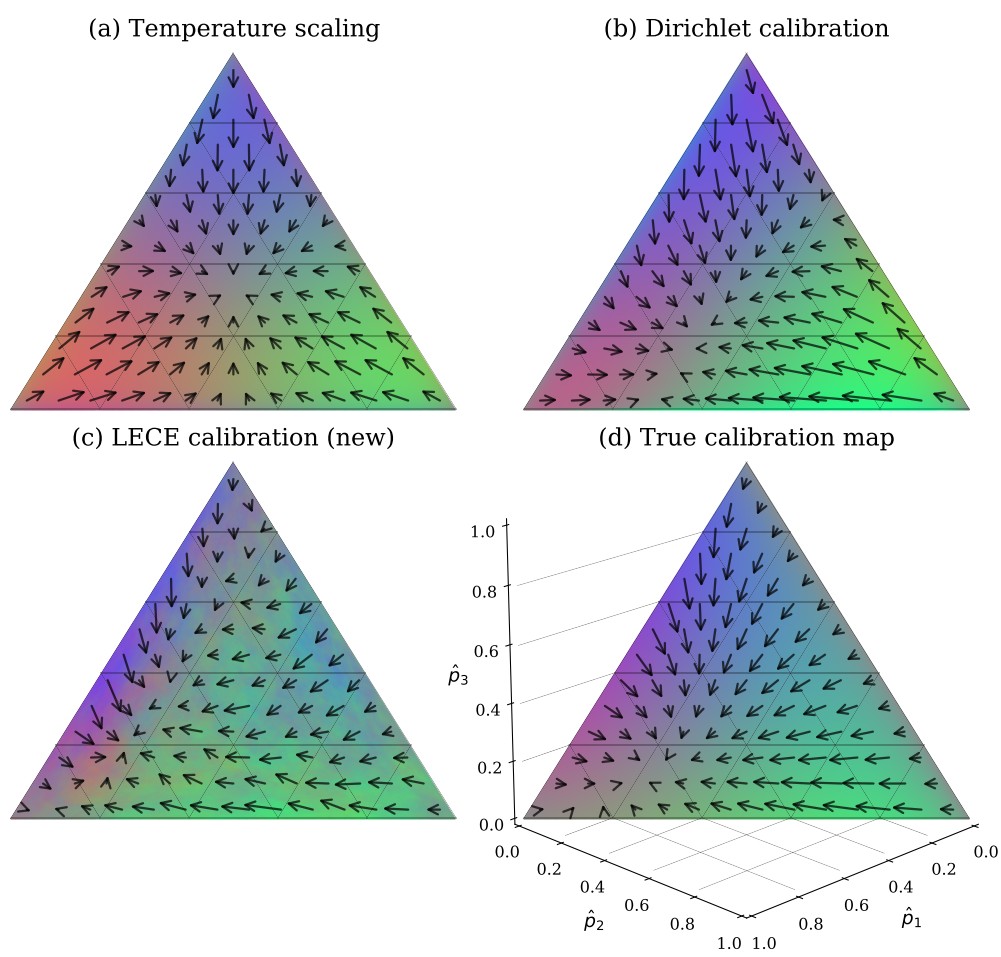

(a) Temperature scaling

(b) Dirichlet calibration

(c) LECE calibration (new)

(d) True calibration map

Figure 2: Illustrative example of calibration maps produced by different post-hoc calibration methods on a synthetic calibration task (a, b, c). The goal of the methods is to learn the true calibration map (d). Temperature scaling (a) is limited by its symmetric calibration map. Dirichlet calibration (b) performs well, but is held back by its parametric family: it fails to imitate the calibration arrows of the true calibration map (d) for small values of $\hat{p}_1$. LECE calibration (c) manages to learn a transformation very similar to the true calibration map (d).

errors (CE) as the true calibration function is known (see Section 2.3 for details): CE is measured with $\alpha = 1$ and the expectation is replaced with empirical test set average. The last column in Table 1 depicts the results when applying the true calibration map — the theoretical best result the methods could achieve.

First, when comparing the two histogram binnings, it can be seen that the evaluation metrics in Table 1 confirm the visual intuition available from Figure 1a and Figure 1b: the new proposed LECE assumption significantly outperforms the classical LECD assumption. Second, as seen in Figure 2a, temperature scaling is clearly not sufficient for the task as it is limited to a symmetrical transformation. Third, Dirichlet calibration, one of the strongest existing competitors learns a result close to the true calibration map, but is held back by its parametric family: note its bad performance for values close to 0 for $\hat{p}_1$ as the learned calibration arrows there are not similar to the arrows of the true calibration map. Overall, the proposed LECE calibration method depicted in Figure 2c learns the most similar transformation to the true calibration map.

## 4.2 Real data experiments

The goal of the real data experiments is to see if the proposed method can improve state-of-the-art in practical settings.

### 4.2.1 Datasets and models

CIFAR-10 and CIFAR-100 datasets (Krizhevsky, 2009) are used for the experiments. On both of the datasets, the predictions of convolutional neural networks ResNet-110 (He et al., 2016), ResNet Wide 32 (Zagoruyko & Komodakis, 2016), DenseNet-40 (Huang et al., 2017) are used. The precomputed logits of these three CNN-s were provided by Kull et al. (2019), and the same validation and test set split of the logits was used as in the experiments of Kull et al. (2019) and Rahimi et al. (2020). An overview of the used datasets, classifiers, and dataset sizes is given in Table 2.

### 4.2.2 Compared post-hoc calibration methods

On the real datasets the following methods are compared:

- Uncalibrated predictions (**uncal**) to have a baseline.

- Matrix scaling with ODIR (**MS**); best hyperparameters for the dataset and classifier combinations were provided by the authors of ODIR for matrix scaling (Kull et al., 2019).

- Diagonal subfamily of intra-order preserving functions (**IOP**); best hyperparameters for the dataset and classifier combinations were obtained from the original article (Rahimi et al., 2020).

- Gaussian process calibration (**GP**) applied on logits (Wenger et al., 2020).

- Temperature scaling (**TS**) (Guo et al., 2017).

Table 2: Datasets and model details for the real experiments. The precomputed logits were provided by Kull et al. (2019), and the same validation and test set split of the logits was used as in the experiments of Kull et al. (2019) and Rahimi et al. (2020).

| Dataset | Models | Classes | Dataset size | | |
| | | | Training | Validation | Test |
| --- | --- | --- | --- | --- | --- |
| CIFAR-10 | DenseNet-40 ResNet-110 ResNet Wide 32 | 10 | 45000 | 5000 | 10000 |
| CIFAR-100 | DenseNet-40 ResNet-110 ResNet Wide 32 | 100 | 45000 | 5000 | 10000 |

- Decision calibration (Zhao et al., 2021); trained to achieve decision calibration with respect to all loss functions with 2 decisions; number of trained iterations was determined by looking at the test set (!) Brier score to save computational resources, thus giving a slight unfair advantage to this method; the final output was normalised to sum to 1 which was not done in the original implementation but is inevitable for log-loss evaluation; applied in composition with temperature scaling (**TS+DEC**) as recommended by the original paper.

- Isotonic regression calibration (Zadrozny & Elkan, 2002) applied in a one-vs-rest approach (**IR**) and the same method in composition with temperature scaling (**TS+IR**) as proposed by Zhang et al. (2020). In both cases, the final output is normalized by dividing the output vector with its sum.

- Our proposed method; optimal hyperparameters were found with 10-fold cross-validation grid search optimized by log-loss from neighborhood size proportions $q$ of the training dataset $q = k/n \in \{0.01, 0.02, 0.03, 0.04, 0.05, 0.06, 0.07, 0.1, 0.2, 1.0\}$, and threshold values $t \in \{0, 0.00125, 0.0025, 0.005, 0.01, 0.02, 0.04, 0.05, 0.10, 1.0\}$. Note that including $t = 1$ as a possible value allows the calibration method to learn the identity map. Neighborhood size proportion 0.01 stands for $0.01 \cdot 5000 = 50$ neighbors in the validation set and $0.01 \cdot 4500 = 45$ neighbors in one fold of the 10-fold cross-validation as there are 5000 total data points in the validation set. The reason for using a fixed proportion $q$ instead of a fixed number of neighbors $k$ is to ensure that the neighborhood would cover approximately the same subregion of the probability simplex on data folds of different sizes. LECE calibration is applied in composition with temperature scaling (**TS+LECE**) and without temperature scaling (**LECE**).

For methods applied in composition with temperature scaling, temperature scaling is always applied first.

### 4.2.3 Results

In the following paragraphs the results for confidence ECE, classwise ECE, log-loss, and accuracy are presented.

**Confidence ECE**  Table 3 presents the results for confidence ECE. Both confidence and classwise ECE are measured with 15 equal-sized bins or also known as data dependent bins according to the formulas described in Section 2.4. According to confidence ECE, the best performing methods are GP and our proposed method TS+LECE. Without TS, LECE performs poorly on the real datasets: it is heavily outperformed by every other method. However, when applied in composition with TS, the result of TS is improved for 5 out of 6 cases.

Table 3: Confidence ECE ($\times 10^2$) applied with 15 equal-sized bins according to the formula described in Section 2.4. For details of the applied methods see Section 4.2.2. In every row, the methods are ranked and the rank is displayed as a subindex, the best performing method is also highlighted in bold. On average, the best performing methods are Gaussian process calibration (GP) and our proposed method TS+LECE.

|  |  |  |  | ours |  |  |  |  |  |  | ours |
|---|---|---|---|---|---|---|---|---|---|---|---|
|  |  | uncal | IR | LECE | MS | IOP | GP | TS | TS+DEC | TS+IR | TS+LECE |
| C-10 | DenseNet-40 | $5.49_{10}$ | $1.67_8$ | $3.72_9$ | $0.91_4$ | $0.80_2$ | $0.86_3$ | $0.92_5$ | $1.13_6$ | $1.39_7$ | $\mathbf{0.69_1}$ |
|  | ResNet-110 | $4.75_{10}$ | $1.44_8$ | $3.31_9$ | $0.99_5$ | $0.91_3$ | $0.88_2$ | $0.94_4$ | $1.01_6$ | $1.02_7$ | $\mathbf{0.65_1}$ |
|  | ResNet Wide 32 | $4.48_{10}$ | $1.07_8$ | $2.05_9$ | $0.75_6$ | $0.69_3$ | $\mathbf{0.39_1}$ | $0.69_3$ | $0.76_7$ | $0.69_3$ | $0.60_2$ |
|  | average rank | 10.0 | 8.0 | 9.0 | 5.0 | 2.7 | 2.0 | 4.0 | 6.3 | 5.7 | 1.3 |
| C-100 | DenseNet-40 | $21.16_{10}$ | $4.86_8$ | $10.49_9$ | $1.22_4$ | $3.45_6$ | $0.89_2$ | $\mathbf{0.79_1}$ | $3.85_7$ | $2.12_5$ | $1.18_3$ |
|  | ResNet-110 | $18.48_{10}$ | $5.80_8$ | $6.36_9$ | $2.31_4$ | $2.80_5$ | $\mathbf{1.89_1}$ | $2.13_3$ | $3.40_7$ | $3.17_6$ | $1.99_2$ |
|  | ResNet Wide 32 | $18.78_{10}$ | $5.74_8$ | $15.53_9$ | $1.85_5$ | $1.04_3$ | $0.87_2$ | $1.41_4$ | $3.34_7$ | $2.95_6$ | $\mathbf{0.85_1}$ |
|  | average rank | 10.0 | 8.0 | 9.0 | 4.3 | 4.7 | 1.7 | 2.7 | 7.0 | 5.7 | 2.0 |

Table 4: Classwise ECE ($\times 10^2$) applied with 15 equal-sized bins according to the formula described in Section 2.4. For details of the applied methods see Section 4.2.2. In every row, the methods are ranked and the rank is displayed as a subindex, the best performing method is also highlighted in bold. On average, the best performing methods are matrix scaling (MS) and our proposed method TS+LECE. Row CIFAR-10 ResNet Wide 32 illustratres the limitations of symmetrical methods, where all symmetrical methods (IOP, GP, TS) offer minimal improvements over uncalibrated predictions.

| | | uncal | IR | ours
LECE | MS | IOP | GP | TS | TS+DEC | TS+IR | ours
TS+LECE |
|---|---|---|---|---|---|---|---|---|---|---|---|
| C-10 | DenseNet-40 | $0.445_{10}$ | $0.280_7$ | $0.411_9$ | $\mathbf{0.214_1}$ | $0.251_3$ | $0.259_5$ | $0.255_4$ | $0.291_8$ | $0.264_6$ | $0.234_2$ |
| | ResNet-110 | $0.358_{10}$ | $0.260_8$ | $0.297_9$ | $\mathbf{0.180_1}$ | $0.212_2$ | $0.215_3$ | $0.216_4$ | $0.248_7$ | $0.234_6$ | $0.216_4$ |
| | ResNet Wide 32 | $0.496_{10}$ | $0.319_6$ | $0.292_5$ | $\mathbf{0.181_1}$ | $0.452_9$ | $0.440_7$ | $0.446_8$ | $0.282_4$ | $0.246_2$ | $0.246_2$ |
| | average rank | 10.0 | 7.0 | 7.7 | 1.0 | 4.7 | 5.0 | 5.3 | 6.3 | 4.7 | 2.7 |
| C-100 | DenseNet-40 | $0.130_8$ | $0.117_7$ | $0.206_{10}$ | $0.095_2$ | $0.110_6$ | $0.102_4$ | $0.102_4$ | $0.139_9$ | $0.095_2$ | $\mathbf{0.094_1}$ |
| | ResNet-110 | $0.132_8$ | $0.119_7$ | $0.159_{10}$ | $0.110_5$ | $0.096_4$ | $\mathbf{0.092_1}$ | $0.094_3$ | $0.135_9$ | $0.114_6$ | $\mathbf{0.092_1}$ |
| | ResNet Wide 32 | $0.124_8$ | $0.122_7$ | $0.136_{10}$ | $0.094_2$ | $0.105_5$ | $0.105_5$ | $0.103_4$ | $0.124_8$ | $0.096_3$ | $\mathbf{0.089_1}$ |
| | average rank | 8.0 | 7.0 | 10.0 | 3.0 | 5.0 | 3.3 | 3.7 | 8.7 | 3.7 | 1.0 |

**Classwise ECE**  Table 4 presents the results for classwise ECE. Note that classwise ECE values tend to be a lot smaller than confidence ECE: this is due to the fact that most predictions coming from the softmax function are tiny and wash out the ECE score (Nixon et al., 2019). On CIFAR-10, matrix scaling is clearly the best performing method. On CIFAR-100 our proposed TS+LECE performs best but only marginally: many other methods offer similar scores. The results on CIFAR-10 ResNet Wide 32 expose the limitations of symmetrical methods that perform the same transformation for all the classes. For that case, IOP, GP, and TS all fail and produce bad classwise ECE scores around 0.440 which is only slightly lower than the uncalibrated result 0.496. Methods not limited by symmetry offer substantially better results, all producing scores less than 0.320 with matrix scaling even reaching as low as 0.181. Without TS, LECE performs again poorly: for CIFAR-100 it even worsens the result of uncalibrated predictions. However, similarly to confidence ECE, when applied in composition with TS, it offers consistent improvements over TS.

**Log-loss**  Table 5 displays the results for log-loss. The best method according log-loss is clearly matrix scaling, being ranked first every time. The second best method is our proposed TS+LECE. The limitations of symmetrical methods can be seen according to log-loss as well: on CIFAR-10 ResNet Wide 32, IOP, GP, and TS perform worse than MS and TS+LECE. Without TS, LECE again performs poorly, but in composition with TS, it consistently offers improvements over TS.

Table 5: Log-loss. For details of the applied methods see Section 4.2.2. In every row, the methods are ranked and the rank is displayed as a subindex, the best performing method is also highlighted in bold. On average, the best performing method is matrix scaling (MS) followed by our proposed TS+LECE.

| | | uncal | IR | ours
LECE | MS | IOP | GP | TS | TS+DEC | TS+IR | ours
TS+LECE |
|---|---|---|---|---|---|---|---|---|---|---|---|
| C-10 | DenseNet-40 | $0.428_{10}$ | $0.268_8$ | $0.310_9$ | $\mathbf{0.222_1}$ | $0.225_3$ | $0.226_5$ | $0.225_3$ | $0.266_7$ | $0.261_6$ | $0.223_2$ |
| | ResNet-110 | $0.358_{10}$ | $0.250_7$ | $0.267_9$ | $\mathbf{0.204_1}$ | $0.208_4$ | $0.206_2$ | $0.209_5$ | $0.232_6$ | $0.254_8$ | $0.206_2$ |
| | ResNet Wide 32 | $0.382_{10}$ | $0.214_6$ | $0.264_9$ | $\mathbf{0.182_1}$ | $0.192_5$ | $0.191_3$ | $0.191_3$ | $0.234_8$ | $0.219_7$ | $0.185_2$ |
| | average rank | 10.0 | 7.0 | 9.0 | 1.0 | 4.0 | 3.3 | 3.7 | 7.0 | 7.0 | 2.0 |
| C-100 | DenseNet-40 | $2.017_{10}$ | $1.367_7$ | $1.739_9$ | $\mathbf{1.047_1}$ | $1.066_5$ | $1.056_3$ | $1.057_4$ | $1.338_6$ | $1.442_8$ | $1.054_2$ |
| | ResNet-110 | $1.694_{10}$ | $1.619_9$ | $1.498_7$ | $\mathbf{1.073_1}$ | $1.105_5$ | $1.082_2$ | $1.092_4$ | $1.453_6$ | $1.591_8$ | $1.085_3$ |
| | ResNet Wide 32 | $1.802_{10}$ | $1.365_7$ | $1.576_9$ | $\mathbf{0.931_1}$ | $0.945_4$ | $0.939_3$ | $0.945_4$ | $1.284_6$ | $1.401_8$ | $0.937_2$ |
| | average rank | 10.0 | 7.7 | 8.3 | 1.0 | 4.7 | 2.7 | 4.0 | 6.0 | 8.0 | 2.3 |

Table 6: Classifier accuracy. For details of the applied methods see Section 4.2.2. In every row, the methods are ranked and the rank is displayed as a subindex, the best performing method is also highlighted in bold. There are no large differences between the methods. Only notable value is isotonic regression combined with temperature scaling (TS+IR) on CIFAR-100 ResNet-110, where it reduces accuracy from 0.715 to 0.710.

| | | | | ours | | | | | | | ours |
| --- | --- | --- | --- | --- | --- | --- | --- | --- | --- | --- | --- |
| | | uncal | IR | LECE | MS | IOP | GP | TS | TS+DEC | TS+IR | TS+LECE |
| C-10 | DenseNet-40 | $0.924_3$ | $0.924_3$ | $0.924_3$ | $\mathbf{0.925_1}$ | $0.924_3$ | $0.924_3$ | $0.924_3$ | $0.924_3$ | $0.924_3$ | $\mathbf{0.925_1}$ |
| | ResNet-110 | $\mathbf{0.936_1}$ | $\mathbf{0.936_1}$ | $\mathbf{0.936_1}$ | $\mathbf{0.936_1}$ | $\mathbf{0.936_1}$ | $0.935_9$ | $\mathbf{0.936_1}$ | $\mathbf{0.936_1}$ | $0.935_9$ | $\mathbf{0.936_1}$ |
| | ResNet Wide 32 | $0.939_7$ | $0.940_5$ | $0.940_5$ | $\mathbf{0.942_1}$ | $0.939_7$ | $0.939_7$ | $0.939_7$ | $0.941_3$ | $0.941_3$ | $\mathbf{0.942_1}$ |
| | average rank | 3.7 | 3.0 | 3.0 | 1.0 | 3.7 | 6.3 | 3.7 | 2.3 | 5.0 | 1.0 |
| C-100 | DenseNet-40 | $0.700_4$ | $0.700_4$ | $0.700_4$ | $\mathbf{0.704_1}$ | $0.700_4$ | $0.700_4$ | $0.700_4$ | $0.702_3$ | $0.697_{10}$ | $\mathbf{0.704_1}$ |
| | ResNet-110 | $0.715_3$ | $0.712_9$ | $0.715_3$ | $0.715_3$ | $0.715_3$ | $0.715_3$ | $0.715_3$ | $\mathbf{0.716_1}$ | $0.710_{10}$ | $\mathbf{0.716_1}$ |
| | ResNet Wide 32 | $0.738_4$ | $0.738_4$ | $0.738_4$ | $\mathbf{0.740_1}$ | $0.738_4$ | $0.738_4$ | $0.738_4$ | $\mathbf{0.740_1}$ | $0.736_{10}$ | $\mathbf{0.740_1}$ |
| | average rank | 3.7 | 5.7 | 3.7 | 1.7 | 3.7 | 3.7 | 3.7 | 1.7 | 10.0 | 1.0 |

**Accuracy**   Table 6 presents accuracies of the methods on the test set. None of the methods offer substantial improvements in accuracy, nor do any of the methods have considerable detrimental effects. In general, the methods perform very similarly. The only notable value is TS+IR on CIFAR-100 ResNet-110, where it reduces accuracy of the uncalibrated classifier from 0.715 to 0.710.

### 4.2.4   Ablation study

To understand the importance of the neighborhood function used in the LECE calibration algorithm, and to compare LECE calibration with LECD calibration, we repeat the real data experiments for the following methods

- LECE calibration with Euclidean distance instead of Kullback-Leibler divergence ($\mathbf{LECE}_{euc}$ and $\mathbf{TS{+}LECE}_{euc}$),

- LECD calibration as described in Section 3.2 — otherwise the same method as LECE calibration but using the LECD assumption instead of the LECE assumption ($\mathbf{LECD}$ and $\mathbf{TS{+}LECD}$).

The method parameters were chosen with 10-fold cross-validation from the same parameter sets as for LECE calibration described in Section 4.2.2. Table 7 presents the results for confidence ECE, Table 8 the results for

Table 7: Ablation study, confidence ECE ($\times 10^2$) applied with 15 equal-sized bins according to the formula described in Section 2.4. For details of the applied methods see Section 4.2.4. In every row, the methods are ranked and the rank is displayed as a subindex, the best performing method is also highlighted in bold. On average, the best performing method is TS+LECE.

| | | LECE | $LECE_{euc}$ | LECD | TS+LECE | $TS{+}LECE_{euc}$ | TS+LECD |
| --- | --- | --- | --- | --- | --- | --- | --- |
| C-10 | DenseNet-40 | $3.72_6$ | $3.30_4$ | $3.43_5$ | $\mathbf{0.69_1}$ | $0.76_2$ | $1.19_3$ |
| | ResNet-110 | $3.31_6$ | $3.20_5$ | $2.69_4$ | $\mathbf{0.65_1}$ | $0.66_2$ | $0.94_3$ |
| | ResNet Wide 32 | $2.05_5$ | $2.93_6$ | $1.93_4$ | $0.60_3$ | $0.53_2$ | $\mathbf{0.50_1}$ |
| | average rank | 5.7 | 5.0 | 4.3 | 1.7 | 2.0 | 2.3 |
| C-100 | DenseNet-40 | $10.49_5$ | $11.16_6$ | $5.97_4$ | $1.18_2$ | $1.21_3$ | $\mathbf{0.79_1}$ |
| | ResNet-110 | $6.36_5$ | $12.02_6$ | $3.53_4$ | $1.99_2$ | $2.03_3$ | $\mathbf{1.71_1}$ |
| | ResNet Wide 32 | $15.53_6$ | $14.21_5$ | $5.76_4$ | $\mathbf{0.85_1}$ | $1.16_2$ | $1.41_3$ |
| | average rank | 5.3 | 5.7 | 4.0 | 1.7 | 2.7 | 1.7 |

Table 8: Ablation study, classwise ECE ($\times 10^2$) applied with 15 equal-sized bins according to the formula described in Section 2.4. For details of the applied methods see Section 4.2.4. In every row, the methods are ranked and the rank is displayed as a subindex, the best performing method is also highlighted in bold. On average, the best performing method is TS+LECE.

|  |  | LECE | $\text{LECE}_{euc}$ | LECD | TS+LECE | $\text{TS+LECE}_{euc}$ | TS+LECD |
|---|---|---|---|---|---|---|---|
| C-10 | DenseNet-40 | $0.411_4$ | $0.670_6$ | $0.453_5$ | $\mathbf{0.234_1}$ | $0.238_2$ | $0.267_3$ |
|  | ResNet-110 | $0.297_4$ | $0.557_6$ | $0.335_5$ | $0.216_2$ | $\mathbf{0.202_1}$ | $0.216_2$ |
|  | ResNet Wide 32 | $0.292_5$ | $0.732_6$ | $0.285_4$ | $\mathbf{0.246_1}$ | $0.274_2$ | $0.277_3$ |
|  | average rank | 4.3 | 6.0 | 4.7 | 1.3 | 1.7 | 2.7 |
| C-100 | DenseNet-40 | $0.251_5$ | $0.336_6$ | $0.172_4$ | $\mathbf{0.094_1}$ | $0.096_2$ | $0.102_3$ |
|  | ResNet-110 | $0.159_5$ | $0.269_6$ | $0.123_4$ | $\mathbf{0.092_1}$ | $0.095_3$ | $\mathbf{0.092_1}$ |
|  | ResNet Wide 32 | $0.137_5$ | $0.332_6$ | $0.126_4$ | $\mathbf{0.089_1}$ | $0.095_2$ | $0.103_3$ |
|  | average rank | 5.0 | 6.0 | 4.0 | 1.0 | 2.3 | 2.3 |

Table 9: Ablation study, log-loss. For details of the applied methods see Section 4.2.4. In every row, the methods are ranked and the rank is displayed as a subindex, the best performing method is also highlighted in bold. The best performing method is TS+LECE.

|  |  | LECE | $\text{LECE}_{euc}$ | LECD | TS+LECE | $\text{TS+LECE}_{euc}$ | TS+LECD |
|---|---|---|---|---|---|---|---|
| C-10 | DenseNet-40 | $0.310_5$ | $0.305_4$ | $0.319_6$ | $\mathbf{0.223_1}$ | $0.224_2$ | $0.226_3$ |
|  | ResNet-110 | $0.267_5$ | $0.265_4$ | $0.279_6$ | $\mathbf{0.206_1}$ | $\mathbf{0.206_1}$ | $0.209_3$ |
|  | ResNet Wide 32 | $0.264_5$ | $0.260_4$ | $0.272_6$ | $\mathbf{0.185_1}$ | $0.187_2$ | $0.188_3$ |
|  | average rank | 5.0 | 4.0 | 6.0 | 1.0 | 1.7 | 3.0 |
| C-100 | DenseNet-40 | $1.739_6$ | $1.681_5$ | $1.437_4$ | $\mathbf{1.054_1}$ | $\mathbf{1.054_1}$ | $1.057_3$ |
|  | ResNet-110 | $1.498_5$ | $1.516_6$ | $1.306_4$ | $\mathbf{1.085_1}$ | $1.087_2$ | $1.092_3$ |
|  | ResNet Wide 32 | $1.576_6$ | $1.544_5$ | $1.272_4$ | $\mathbf{0.937_1}$ | $0.941_2$ | $0.945_3$ |
|  | average rank | 5.7 | 5.3 | 4.0 | 1.0 | 1.7 | 3.0 |

classwise ECE, and Table 9 the results for log-loss. The three tables are discussed in unison, as the methods are ranked similarly across the tables, and the key conclusions to be made from the tables are the same.

The best method on average in all the tables is TS+LECE. TS+LECE outperforms its derivative with Euclidean distance $\text{TS+LECE}_{euc}$ in almost all cases with a few exceptions: it loses once in confidence ECE, once in classwise ECE, and is tied twice in log-loss. TS+LECE outperforms TS+LECD as well: it performs better in 13 cases, is tied in 2, and loses in 3 out of the 18 total rows in the three tables.

When comparing the methods with and without TS, it can be seen that TS is crucial for all of them. Adding the composition with TS improves the result in all cases and by a very large margin. For cases without TS, LECE and $\text{LECE}_{euc}$ perform otherwise similarly but for classwise ECE the method $\text{LECE}_{euc}$ fails. Therefore, similary to the LECE methods applied in composition with TS, Kullback-Leibler divergence can be concluded to perform better than Euclidean distance for the non-compositional case as well. Comparing LECE with LECD, LECE seems to perform better for CIFAR-10 but LECD for CIFAR-100. Yet, as the compositional TS+LECE methods heavily outperformed the non-compositonal LECE methods, the final conclusion would still be that LECE assumption is better than the LECD assumption.

Table 10 shows the optimal neighborhood proportion parameter $q$ chosen by the 10-fold cross-validation for the methods. Many different values are represented, starting from 0.01 corresponding to $0.01 \cdot 5000 = 50$

Table 10: Optimal neighborhood proportion $q$ chosen by cross-validation. Commonly chosen values are between 0.01 and 0.1 (corresponding respectively to 50 and 500 neighbors). In some cases the value 1.0 is also chosen corresponding to the whole 5000 validation data set points. For TS+LECE, the neighborhood proportion remains in range between 0.01 to 0.04 corresponding to 50 to 200 neighbors.

| | | LECE | LECE$_{euc}$ | LECD | TS+LECE | TS+LECE$_{euc}$ | TS+LECD |
|---|---|---|---|---|---|---|---|
| C-10 | DenseNet-40 | 0.04 | 0.1 | 0.04 | 0.04 | 0.03 | 0.07 |
| | ResNet-110 | 0.06 | 0.1 | 1.0 | 0.03 | 0.02 | 0.01 |
| | ResNet Wide 32 | 0.02 | 0.1 | 0.02 | 0.01 | 0.01 | 0.01 |
| C-100 | DenseNet-40 | 0.05 | 0.03 | 1.0 | 0.01 | 0.02 | 0.01 |
| | ResNet-110 | 0.01 | 0.05 | 1.0 | 0.04 | 0.03 | 0.01 |
| | ResNet Wide 32 | 1.0 | 0.03 | 1.0 | 0.04 | 0.05 | 0.01 |

Table 11: Optimal threshold $t$ chosen by cross-validation. For methods applied without TS, values close to 0 are chosen, corresponding to no thresholding. For methods applied in composition with TS, the chosen threshold values are usually small: between 0.0025 and 0.02. In three cases, TS+LECD has chosen $t = 1.0$ — meaning it has learned the identity transformation.

| | | LECE | LECE$_{euc}$ | LECD | TS+LECE | TS+LECE$_{euc}$ | TS+LECD |
|---|---|---|---|---|---|---|---|
| C-10 | DenseNet-40 | 0 | 0 | 0 | 0.0025 | 0.0025 | 0.1 |
| | ResNet-110 | 0 | 0 | 0.05 | 0.01 | 0.01 | 1.0 |
| | ResNet Wide 32 | 0 | 0 | 0 | 0.01 | 0.02 | 0.05 |
| C-100 | DenseNet-40 | 0 | 0 | 0.005 | 0.02 | 0.02 | 1.0 |
| | ResNet-110 | 0 | 0 | 0.005 | 0.01 | 0.02 | 0.1 |
| | ResNet Wide 32 | 0 | 0 | 0.005 | 0.005 | 0.005 | 1.0 |

neighbors, up to 1.0, corresponding to the whole validation dataset of 5000 neighbors. For TS+LECE, the neighborhood proportion remains in range between 0.01 to 0.04 corresponding to 50 to 200 neighbors.

Table 11 shows the optimal threshold parameter chosen by the 10-fold cross-validation. For methods applied without TS, 0 seems to be a good value as it was chosen by LECE and LECE$_{euc}$ in all cases. For methods applied in composition with TS, the chosen threshold values remain usually small: between 0.0025 and 0.02. TS+LECD is an exception as it uses larger threshold values. In three cases out of six it even picked $t = 1.0$, meaning it learned the identity map and kept the result of TS.

The running times of LECE and LECD calibration are discussed in Appendix A.3.

### 4.2.5 Discussion

Overall, from the experiments it can be concluded that while many strong methods exist, our proposed TS+LECE can still offer improvements. Symmetrical methods IOP, GP, and TS perform generally well, but can fail for problems where asymmetrical transformations are needed as shown by classwise ECE and log-loss on CIFAR-10 ResNet Wide 32. Matrix scaling could be considered the best method according to classwise ECE and log-loss, but according to confidence ECE it is clearly outperformed by other methods. TS+LECE avoids the problems of symmetrical methods and offers the best confidence ECE, being second best in classwise ECE and log-loss.

Another aspect to notice from the experiments, is the behaviour of LECE with the number of classes. On the illustrational synthetic task with 3 classes the method performs very well without TS. However, on the real tasks with 10 and 100 classes, the method fails unless it is used on top of TS. This is due to problems arising from the curse of dimensionality inherent to the proposed non-parametric approach. In higher dimensions, the probability space is more sparsely populated as every data point starts to become approximately equally

distant from every other data point. Because of this, the neighborhood sizes grow and the applied assumption of locally equal calibration errors becomes very bold, lowering the effectiveness of the method. However, applying LECE on top of TS makes the assumption to be more realistic, as the calibration errors are smaller after TS and hence the errors of neighbors are more similar, as required by the LECE assumption.

## 5 Conclusion

This work explored the field of post-hoc calibration methods for multi-class classifiers. Two assumptions about the true calibration map were formalized that have been previously used for creating estimators, but despite this have never been clearly stated: assuming locally equal calibration errors (LECE) and assuming locally equal class distributions (LECD). Based on the more reasonable of the assumptions, a non-parametric calibration method was proposed — LECE calibration. The used assumption states that the calibration error of a data point can be modeled by the calibration errors of its neighbors. This results in using the average difference of predictions and labels in a close neighborhood to estimate the calibration error of a prediction. Based on the definition of calibration, the found calibration error estimate is then subtracted from the prediction to reach a calibrated prediction.

Experiments were carried out on three convolutional neural networks trained on CIFAR-10 and CIFAR-100 to compare the proposed method with its competitors. The experimental results on real data showed that the proposed method alone is clearly not competitive for cases with many classes and a limited validation dataset due to problems arising from the curse of dimensionality. However, when applying the proposed method in composition with temperature scaling, it tops the state-of-the-art in confidence ECE and is close to the best according to classwise ECE and log-loss.

For future work, the limitations of the proposed approach could be studied more thoroughly. How does improvement in calibration depend on the number of classes in the dataset; on the validation dataset size; or on the distribution of the predictions? In addition, the composition of different calibration methods could be studied further as this work and several previous (Zhang et al., 2020; Zhao et al., 2021) have shown the possible benefits.

### Acknowledgments

This work was supported by the European Social Fund via IT Academy programme, Estonian Research Council grant PRG1604, and by the Estonian Centre of Excellence in IT (EXCITE), funded by the European Regional Development Fund.

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

# A  Appendix

## A.1  Example of different calibration definitions

As a toy example to illustrate the different definitions of calibration, consider a 3-class classifier for which $\hat{\boldsymbol{P}}$ can take two equally likely values

$$\hat{\boldsymbol{P}} = (0.6, 0.3, 0.1) \text{ or } \hat{\boldsymbol{P}} = (0.6, 0.2, 0.2).$$

Now suppose that the corresponding expected label values for these predictions are

$$\mathbb{E}_{\boldsymbol{Y}}\left[\boldsymbol{Y}|\hat{\boldsymbol{P}} = (0.6, 0.3, 0.1)\right] = (0.5, 0.3, 0.2) \text{ and}$$

$$\mathbb{E}_{\boldsymbol{Y}}\left[\boldsymbol{Y}|\hat{\boldsymbol{P}} = (0.6, 0.2, 0.2)\right] = (0.7, 0.2, 0.1).$$

Such a classifier would not be multi-class calibrated as

$$\mathbb{E}_{\boldsymbol{Y}}\left[\boldsymbol{Y}|\hat{\boldsymbol{P}} = (0.6, 0.3, 0.1)\right] \neq (0.6, 0.3, 0.1), \text{ and also}$$

$$\mathbb{E}_{\boldsymbol{Y}}\left[\boldsymbol{Y}|\hat{\boldsymbol{P}} = (0.6, 0.2, 0.2)\right] \neq (0.6, 0.2, 0.2).$$

The classifier would also not be classwise calibrated as

$$\mathbb{E}_{\boldsymbol{Y}}\left[Y_3|\hat{P}_3 = 0.1\right] \neq 0.1, \text{ and also}$$

$$\mathbb{E}_{\boldsymbol{Y}}\left[Y_3|\hat{P}_3 = 0.2\right] \neq 0.2.$$

However, the classifier would be confidence calibrated as

$$\mathbb{E}_{\boldsymbol{Y}}\left[Y_{argmax\ \hat{\boldsymbol{P}}}|max\ \hat{\boldsymbol{P}} = 0.6\right] = 0.5 \cdot 0.5 + 0.5 \cdot 0.7 = 0.6.$$

### A.2 Computational and memory complexity of LECE calibration

The complete pseudocode of LECE calibration with thresholding was presented in Algorithm 1. The LECE calibration method is essentially a variation of the $k$-nearest-neighbors algorithm and its exact memory and computational complexity depends on the implementation. The LECE calibration method does not need any training, but has heavy computational and memory complexity during inference time. On a validation set with size $x$, a test set with size $y$, $m$ classes, and $k$ neighbors, the total computational complexity of our implementation is $O(m \cdot x \cdot y)$ and it is caused by line 1 of Algorithm 1, which needs $O(m \cdot x)$ calculations for a single test set data point. The memory complexity of our implementation is $O(x \cdot m \cdot b + y \cdot m)$, where $b \leq y$ is a batch size parameter. The memory complexity $O(x \cdot m \cdot b)$ is caused by line 1 of the algorithm, where the distances are calculated with matrix operations applied on test set batches of size $b$. The $O(y \cdot m)$ is also needed as the test dataset has to be kept in memory during inference time.

### A.3 Running times of LECE calibration

The LECE method requires no time for training but considerable time for inference as discussed in Appendix A.2. The real data experiments were implemented in Python and run on a machine with 16 GBs of RAM and a CPU with clock speed 3.7 GHz. For CIFAR-10 DenseNet-40, LECE calibration with the best hyperparameters reported in Table 10 and Table 11 took 4.7s to calibrate the 10000 test set data points given the 5000 validation set data points (average running time over 10 runs). For CIFAR-100 DenseNet-40, the average running time over 10 runs was 34.2s.

By far the most computationally expensive part of the LECE calibration method is the calculation of distances in line 1 of Algorithm 1. For CIFAR-10, line 1 accounted for 85% of the running time (4.0s of 4.7s), and for CIFAR-100 it was even 97% (33.2s of 34.2s). The second most computationally expensive part of the algorithm is finding the $k$ closest neighbors in line 2 of Algorithm 1. For CIFAR-10, line 2 accounted for 13% of total running time, and for CIFAR-100 it was 2%.

The running times for the LECD calibration method were very similar to LECE; as were the running times for TS+LECE and TS+LECD as temperature scaling requires only a fraction of a second for training and evaluation. For example, TS+LECE took 4.9s to train the calibration method and evaluate the 10000 test set points on CIFAR-10 DenseNet-40; and for CIFAR-100 DenseNet-40 it was 34.8s (average of 10 runs). The running times for ResNet-110 and ResNet Wide-32 were similar to DenseNet-40.

