# OpenReview forum: "Assuming Locally Equal Calibration Errors for Non-Parametric Multiclass Calibration"
_TMLR — Accepted by TMLR_

### Review · Reviewer_eGb8 · 2023-03-21

**Summary Of Contributions:**

This work proposes a new multi-class post-hoc non-parametric calibration method. The method is based on the idea that the calibration error is approximately equal in a small neighbourhood of the simplex ("LECE assumption"), and consists of subtracting the uncalibrated classifier by the averaged calibration error in a neighbourhood on the classifier. The algorithm therefore depends on a choice of distance on the simplex, and has one hyperparameter that is chosen by cross-validation: the size of the neighbourhood.

Most of the results consist of benchmarking the algorithm against popular calibration methods (TS, MS, IOP, etc.) and across different architectures (DenseNet-40, ResNet-110, etc.) and datasets, both synthetic and real (CIFAR10, CIFAR100). By itself, the proposed method is outperformed in most of the experiments. However, when combined with temperature scaling, it consistently outperforms all the other benchmarks.

**Audience:**

Yes

**Broader Impact Concerns:**

In my reading there are no broader impact concerns for this work.

**Claims And Evidence:**

Yes

**Requested Changes:**

**Major**:
- Include error bars / standard deviation in every experiment.
- Add a discussion on running time and how much validation data is used for cross-validation / TS.

**Minor**:
- Add a pseudo-code for the proposed algorithm and the coupled TS version.

**Strengths And Weaknesses:**

- **[C1]**:
> *A probabilistic classifier is considered calibrated if it outputs probabilities that are in correspondence
with empirical class proportions.*

Being pedantic, this sentence in the abstract is at odds with the definition of calibration given in Section 2.2, where the calibration is defined as an expectation over the population. I agree that in practice this is approximated by an empirical average over the validation set, but technically these two are different.

- **[C2]**:
> *The problem of over-confident predictions is especially common for modern deep neural networks (Guo et al., 2017; Lakshminarayanan et al., 2017). Distorted output probabilities are also characteristic of many classical machine learning methods such as naive Bayes or decision trees (Niculescu-Mizil & Caruana, 2005; Domingos & Pazzani, 1996).*

Recent theoretical progress (Bai et al. 2021; Clarté et al. 2022a,b) have shown that over-confidence is widespread in models as simple as high-dimensional logistic regression. I think it is important to mention this since there is a widespread belief in the literature that overconfidence is a specific problem to deep neural networks.

- **[C3]**: Assumptions LECD and LECE in Page 6 translate to assumptions on the underlying data distribution / target classifier. More precisely, if data is generated from a target classifier $f_{\star}(x) = \mathbb{P}(Y=1|X=x)$ what do these assumptions intuitively imply on the functional form of $f_{\star}$? For instance, would they be satisfied by a logit or probit model?

- **[C4]**: Why don't the authors report the standard deviation over the independent runs in Table 1-4? This is not a good practice, specially when the difference between the comparison is (for certain methods) on the third digit.

- **[C5]**: Since this is a numerical paper based on comparing very different algorithms, I miss a discussion on running times and sample complexity. Are (at least the order of) the running times similar? Both the proposed method and TS require cross-validation. Does this requires two validation sets (one for TS, one for the hyperparameter) or both are done together? More generally, does all methods use the same quantity of validation data?

- **[C6]**: It would be nice to have a pseudo-code for LECE and LECE+TS. This is sort of done in the list in Page 9, but a pseudo-code with input, steps and output clearly stated could help better the implementation and understanding the points above.

- **[C7]**: I find curious that comparing LECE to TS and to TS+LECE, in most cases the highest decrease in the overconfidence often comes from TS itself, which remains a simple and easy to implement metho. What are your thoughts on that?

**References**

[[Bai et al. 2021]](https://proceedings.mlr.press/v139/bai21c.html) Yu Bai, Song Mei, Huan Wang, Caiming Xiong. *Don’t Just Blame Over-parametrization for Over-confidence: Theoretical Analysis of Calibration in Binary Classification*. Proceedings of the 38th International Conference on Machine Learning, PMLR 139:566-576, 2021.

[[Clarte 2022a]](https://arxiv.org/abs/2202.03295) Lucas Clarté, Bruno Loureiro, Florent Krzakala, Lenka Zdeborová. *Theoretical characterization of uncertainty in high-dimensional linear classification*. arXiv: 2202.03295

[[Clarte 2022b]](https://arxiv.org/abs/2210.12760) Lucas Clarté, Bruno Loureiro, Florent Krzakala, Lenka Zdeborová. *A study of uncertainty quantification in overparametrized high-dimensional models*. arXiv: 2210.12760

---

> ### Author Response · Authors · 2023-04-12
> **Response to Reviewer eGb8 (part 1)**
>
> Thank you for the careful and thorough reading and useful feedback. We have taken your feedback into account and uploaded a new version of the paper with changes highlighted in blue. Below are our responses to your comments.
>
> **[C1] ...Being pedantic, this sentence in the abstract is at odds with the definition of calibration given in Section 2.2, where the calibration is defined as an expectation over the population ...**
>
> We agree that the wording was over-simplified. We have now improved this very first sentence in the abstract.
>
> **[C2] ...Recent theoretical progress (Bai et al. 2021; Clarté et al. 2022a,b) have shown that over-confidence is widespread in models as simple as high-dimensional logistic regression.**
>
> Thank you!
> We now also mention the problem of over-confidence in high-dimensional logistic regression, and added references to the recommended 3 articles.
>
> **[C3] ...if data is generated from a target classifier what do these assumptions intuitively imply on the functional form of $\mathbf{f_*}$?...**
>
> The assumptions do not imply much about the functional form of $\mathbf{f_*}(\mathbf{x})=\mathbb{P}(\mathbf{Y}|\mathbf{X}=\mathbf{x})$.
> LECD and LECE assumptions are about the relationship between the outputs $\hat{\mathbf{p}}$ of the learned classifier $\mathbf{f}$ and the true calibration map $\mathbb{P}(\mathbf{Y}|\mathbf{f}(\mathbf{X})=\mathbf{f}(\mathbf{x}))$. For any $\mathbf{f_*}$ (maybe except in some pathological cases), there probably exist classifiers $\mathbf{f}$ for which LECE holds, for which LECD holds, or for which neither of these holds (for some fixed small $\epsilon$).
>
> **[C4] Why don't the authors report the standard deviation over the independent runs in Table 1-4?...**
>
> For the synthetic experiment, we increased the test set size from 10k to 100k and added the standard deviation over the ten seeds to the Table 1.
>
> For the real experiments we use the precomputed logits provided by Kull et al. (2019), and exactly the same validation and test set split of the logits as do the experiments of Kull et al. (2019) and Rahimi et al. (2020).
> In addition, we also use the optimal hyperparameter values of Dirichlet calibration (Kull et al, 2019) and the diagonal subfamily intra-order preserving functions (Rahimi et al., 2020) found in the original articles on the same validation set logits.
> Because of this we have only one single run of the experiments and no standard deviations can be reported.
>
> **[C5] ...I miss a discussion on running times and sample complexity. Are (at least the order of) the running times similar?**
>
> We have added a discussion about the memory and computational complexity of LECE calibration in Section 3.2.
> We have also added a discussion about the running times of LECE in real experiments in Section 4.2.5.
>
> **(C5 continues) Both the proposed method and TS require cross-validation. Does this requires two validation sets (one for TS, one for the hyperparameter) or both are done together?**
>
> Both TS and LECE use the same 5000 validation set data points for method training.
> The compositional method TS+LECE also uses the same 5000 validation set points:
>
> * first, TS uses the 5000 points to learn an optimal temperature parameter;
>
> * second, TS calibration with the learned temperature is applied on the same 5000 validation set points that were used for learning the temperature;
>
> * third, LECE calibration is trained on the same 5000 validation set points, and the optimal hyperparameters are found with cross-validation;
>
> * finally, during inference time, the 10000 test set points are passed through temperature scaling, and then through the LECE calibration method.
>
> Also, note that TS does not require cross-validation to learn the temperature parameter, because the temperature parameter can be optimized directly with log-loss and the BFGS optimization algorithm.
>
> **(C5 continues) More generally, does all methods use the same quantity of validation data?**
>
> Yes, all methods use exactly the same validation set of 5000 data points as detailed in Table 2.
>
> **[C6]: It would be nice to have a pseudo-code for LECE and LECE+TS**
>
> Thank you for this good suggestion!
> We added the pseudo-code for LECE calibration in Section 3.2.
> Note that for TS+LECE, the only difference in the pseudo-code would be that the inputs $\mathbf{\hat{p}}$ and $\mathbf{\hat{p}_1},\dots,\mathbf{\hat{p}_n}$ would be the output of temperature scaling.
> These details are now also mentioned when first referencing the pseudo-code in Section 3.2.

---

> ### Author Response · Authors · 2023-04-12
> **Response to Reviewer eGb8 (part 2)**
>
> **[C7]: I find curious that comparing LECE to TS and to TS+LECE, in most cases the highest decrease in the overconfidence often comes from TS itself, which remains a simple and easy to implement metho. What are your thoughts on that?**
>
> TS assumes that the calibration map is fully symmetric across the classes, with a particular functional form.
> The fact that TS+LECE improves over TS demonstrates that these assumptions are not satisfied.
> In particular, there are two possible reasons for the non-optimality of TS: (1) the true calibration map is not symmetric; (2) the true calibration map is symmetric but has a functional form different from TS.
> Evidence from classise ECE on ResNet Wide 32 CIFAR-100 (Table 4), where symmetric calibration maps are much weaker than non-symmetric ones, suggests that (1) is often true.
>
> **Include error bars / standard deviation in every experiment.**
>
> Answered in question C4.
>
> **Add a discussion on running time and how much validation data is used for cross-validation / TS.**
>
> Answered in question C5.
>
> **Add a pseudo-code for the proposed algorithm and the coupled TS version.**
>
> Answered in question C6.

---

> > ### Comment · Reviewer_eGb8 · 2023-04-13
> > **Post-rebuttal**
> >
> > Thank you for addressing my comments and taking my suggestions into account. Most of my concerns have been addressed in the rebuttal. I have also read the other reviews and believe they provide some nice and constructive suggestions to improve the manuscript.
> >
> > > *The assumptions do not imply much about the functional form of $f_{\star}(x)=\mathbb{P}(y|X=x)$. LECD and LECE assumptions are about the relationship between the outputs $\hat{p}$ of the learned classifier and the true calibration map $\mathbb{P}(y|f(X)=f(x))$. For any $f_{\star}$ (maybe except in some pathological cases), there probably exist classifiers for which LECE holds, for which LECD holds, or for which neither of these holds (for some fixed small $\epsilon$).*
> >
> > I understand that these conditions are not very constraining.
> >
> > However, with my question I just wanted to develop some intuition for what these conditions implicitly imply on the underlying target. For instance, in a probit model where $f_{\star}(x) = \sigma(\theta_{\star}^{\top}x)$ for a for a non-linear activation $\sigma:\mathbb{R}\to (0,1)$ (e.g. erf, sigmoid, etc.), what would these conditions imply on $\theta_{\star}$ and/or $\sigma$? Or put in the complementary way: what are examples of simple statistical models (i.e. not DNNs) that satisfy or violate both, and which satisfy one and not the other?
> >
> > Anyway, this is a minor point and was just for curiosity.

---

> > > ### Author Response · Authors · 2023-04-20
> > > **Re: Post-rebuttal**
> > >
> > > For example, the LECE assumption is satisfied when the classifier is optimal, i.e. $f=f_\star$, and thus there are no calibration errors. In such case, LECE is true for any $\delta,\epsilon>0$. Thus, it does not set any restrictions on the underlying task.
> > >
> > > If the neighbourhood function is the squared Euclidean distance, then for $\delta=\epsilon$ the LECD assumption holds for the optimal classifier $f=f_\star$, not setting any restrictions on the underlying task. If $\epsilon<\delta$, then it does indeed pose some restrictions on the task, but this requires a more detailed analysis as future work.

---

### Review · Reviewer_QVr6 · 2023-03-27

**Summary Of Contributions:**

This work focuses on the **post-hoc calibration of the multi-class probabilistic classifier**. The contributions could be summarized below:

* **1. Formulation of two assumptions** (Section 3.1)

The authors propose two assumptions about the true calibration map:

  (a) Assumption of locally equal calibration errors (LECE)

  (b) Assumption of locally equal class distributions (LECD)

Similar assumptions are implicitly made in existing literature [1, 2, 3, etc] but have never been explicitly stated.

* **2. A detailed visualization comparison of different post-hoc calibration methods**

The authors adopt a well-designed synthetic dataset to compare the performance of a list of post-hoc calibration methods using the calibration map. (Not 100$\%$ sure if this is a novel contribution or not.)

* **3. A non-parametric post-hoc calibration method** (Section 3.2 and Section 4)

The authors proposed LECE calibration, which estimates the calibration error via the averaged difference between the prediction and label, w.r.t. its $k$ nearest neighbors. The synthetic (3-classes) dataset illustrates the effectiveness of the LECE in calibration. On real-world CIFAR datasets, LECE is able to achieve promising results when completed with the temperature scaling method.

**References:**

[1] Obtaining Well Calibrated Probabilities Using Bayesian Binning, AAAI'15.

[2] Calibrating Predictions to Decisions: A Novel Approach to Multi-Class Calibration, NeurIPS'21.

[3] Obtaining Calibrated Probability Estimates from Decision Trees and Naive Bayesian Classifiers, ICML'01.

**Audience:**

Yes

**Broader Impact Concerns:**

There seem to be no ethical concerns about the work.

**Claims And Evidence:**

Yes

**Requested Changes:**

I summarize my concerns, suggestions, and observed typos below, with the order of their appearances in the paper.

* 1. **Suggestion 1:** "Probabilistic classifiers take as input some data and produce as output probability distributions over classes." --> It might be better to say "Probabilistic classifiers take some data as input and produce probability distributions over classes as output."

* 2. **Suggestion 2:** "It is possible that a probabilistic classifier is not well-calibrated and produces distorted probabilities. A classifier is considered to be calibrated if its predicted probabilities are in correspondence with the true class distribution." --> It might be better to switch the order of these two sentences.

* 3. **Suggestion 3:** A toy example for the illustration of three definitions in Section 2.2 could be much better.

* 4. **Suggestion 4:** In Section 2.4, more introduction about proper scoring rules might be better.

* 5. **Notation 1 (Expectation):** on page 6, when talking about expectations, it could be better to what the expectation is with respect to, making the presentation more clear, i.e., $\mathbb{E}_{?} \[\hat{CE}_\{neigh} (\hat{p}) ]$.

* 6. **Suggestion 5:** a brief introduction of the synthetic dataset at the end of page 6 could be much better. Or put the visualization of the assumptions in the experiment section?

* 7. **Question 1:** is the visualization (calibration map) a novel visualization tool for comparisons between post-hoc calibrations?

* 8. **Question 2:** Figure 1 is kind of hard for me to follow. The differences among certain figures (except for (a)) are not quite straightforward to catch. Can authors adopt two sub-figures and explain their differences in more detail (i.e., (d), (e), (f))? Besides, more explanations on settings of the sub-figure are appreciated, i.e., what information is the color conveying. As for the differences between calibration maps, can they be quantified in numeric values?

* 9. **Typo 1:** on page 8, "The LECE assumption outperformed the LECD assumption on the real experiments also". "Also" --> as well.

* 10. **Suggestion 6:** the first sentence of section 3.2 reads weird.

* 11. **Notation 2 ($\hat{p}_{1i}$):** some notations are not well defined/introduced, i.e., in the introduction of KL-divergence on page 9, $\hat{p}_{1i}$ is not well-defined.

* 12. **Notation 3 ($\hat{c}(\hat{p})$):** at the end of paragraph "Thresholding tiny probabilities", the notation is not consistent, i.e., $\hat{\mathbf{c}}(\hat{\mathbf{p}})$ and $\hat{c}(\hat{\mathbf{p}})$.

* 13. **Typo 2:** in the last sentence of the paragraph "Composition with parametric methods", white space is needed in the front.

* 14. **Exp 1:**  can authors try 2 other true calibration functions to see if the performance of LECE is still consistently better than DIR, in the synthetic experiments?

* 15. **Exp 2:** why do authors choose $k=500, t=0$ in the synthetic dataset, and what are the candidate values of $k, t$ in this synthetic experiment?

* 16. **Exp 3:** at the end of section 4.1.3, can the authors explain how the conclusion "the proposed LECE calibration method depicted in Figure 1e learns the most similar transformation to the true calibration map" is made? Is it numerically tested?

* 17. **Exp 4:** in experiments of CIFAR datasets, for TS+LECE, what temperature scaling parameter is adopted? And how is it selected?

* 18. **Exp 5:** for the comparison between TS, and TS+LECE, is TS+LECE always better than TS?

* 19. **Exp 6:** in experiments of CIFAR datasets, while searching for optimal parameters, do the authors have any observations on the pattern of good parameters?

* 20. **Exp 7:** More detailed explanations on why adding LECE requires TS might be needed, for the experiments on CIFAR datasets.

* 21. **Exp 8:** experiment details on how the statistics are calculated (i.e., the evaluation dataset, etc), are needed, for CIFAR experiments.

* 22. **Exp 9:** When looking into the ablation study (Table 7), please correct me if my understanding is incorrect: personally, it seems that the reason why the authors report the performance of TS+LECE rather than others, i.e., $LECE_{\text{euc}}$ or $TS+LECE_{\text{euc}}$ is because of the performance on the dataset for evaluation? For me, I feel like this is kind of cherry-picking. In practice, why do the authors decide on reporting $TS+LECE$ instead of $TS+LECE_{\text{euc}}$ for comparing with other methods?






**Strengths And Weaknesses:**

**Strengths**

* 1. The paper overall is easy to read. The connections between the paper and existing results are well illustrated.

* 2. The "formally" summarized two assumptions are beneficial for the literature.

* 3. The introduced non-parametric calibration works well in synthetic data. When the number of classes is relatively large, combining with temperature scaling can generate promising results.

**Weaknesses**

* 1. **The formulated assumptions are kind of informal**

The two assumptions are not rigorously formulated. A formal illustration could be much better, i.e., instead of mentioning $\hat{p}_1\approx \hat{p}_2\$, a more rigorous saying could be better since these two assumptions are highlighted as contributions. For example, for some distance measure $D$, if $D(\hat{p}_1, \hat{p}_2)<\delta$ for any $\delta\to 0$, ......

* 2. **Experiment results are not fully convincing**

* * 2.1 In the experiments of synthetic datasets, the improvements of LECE (Table 1) are marginal in many settings, i.e., under "accuracy", and "Brier score", the differences between LECE and DIR are negligible. In such cases, is it necessary to give them different ranks? Besides, can authors try 2 other true calibration functions to see if the performance of LECE is still consistently better than DIR? What is more, why do authors choose $k=500, t=0$ in this synthetic dataset, and what are the candidate values of $k, t$ in this synthetic experiment?

* * 2.2 In the experiments of CIFAR datasets, LECE performs too badly without using TS. And it is hard to attribute the nice performance of TS+LECE to LECE. Since in certain cases, TS could achieve fairly well. Adding LECE has no further improvements. A couple of questions are listed in the section **Requested changes**.

* * 2.3 More detailed explanations on why adding LECE requires TS might be needed, for the experiments on CIFAR datasets.

* * 2.4 When looking into the ablation study (Table 7), please correct me if my understanding is incorrect: personally, it seems that the reason why the authors report the performance of TS+LECE rather than others, i.e., $LECE_{\text{euc}}$ or $TS+LECE_{\text{euc}}$ is because of the performance on the dataset for evaluation? For me, I feel like this is kind of cherry-picking. In practice, why do the authors decide on reporting $TS+LECE$ instead of $TS+LECE_{\text{euc}}$ for comparing with other methods?

---

> ### Author Response · Authors · 2023-04-12
> **Response to Reviewer QVr6 (part 1)**
>
> Thank you for the comprehensive review! We have taken your feedback into account and uploaded a new version of the paper with changes highlighted in blue. Below are our responses to your comments.
>
> **Weakness 1: The two assumptions are not rigorously formulated ... instead of mentioning $\hat{p}_1\approx\hat{p}_2$, a more rigorous saying could be better ...**
>
> Thank you for highlighting this important issue! We have now reformulated the assumptions in a formal way, and rewritten significant parts of Section 3.1 accordingly.
>
> **2.1 In the experiments of synthetic datasets, the improvements of LECE (Table 1) are marginal in many settings ... is it necessary to give them different ranks?**
>
> We have improved our wording to move attention to the detail that the synthetic experiment is for illustrational purposes only and removed the ranks from Table 1.
>
> **(2.1 continues) Besides, can authors try 2 other true calibration functions to see if the performance of LECE is still consistently better than DIR?**
>
> The synthetic experiment was never meant as proof of our methods' superiority.
> It was meant merely for the illustrations given in Figure 1 as without a true calibration map (which is never available in practice) one can not illustrate the different calibration methods against the ground truth.
>
> In the comparison of LECE vs DIR it would be possible to construct synthetic datasets to favour either one of the methods.
> In particular, if the true calibration map is chosen to belong to the DIR calibration map family, then DIR would likely be better than LECE.
> In contrast, if the true calibration map is not well approximated by the DIR family, then LECE is likely to be better, given a large enough validation set.
> Thus, in order not to mislead the readers, we opted not to repeat the synthetic experiment with different true calibration functions.
>
> **(2.1 continues) What is more, why do authors choose $k=500,t=0$ in this synthetic dataset, and what are the candidate values of $k,t$ in this synthetic experiment?**
>
> Note that this synthetic calibration task is meant to be purely illustrative of different calibration methods, and therefore the hyperparameters of LECE were manually chosen to show that the method can work well given good hyperparameters.
> The experiments on real data where hyperparameters are tuned with cross-validation show that good hyperparameter values can be found in practice as well.
>
> We have incorporated the above sentences into Section 4.1.2.
> For possible candidate values to use during hyperparameter search, see the parameters tried with cross-validation in Section 4.2.2 in real experiments.
>
> **2.2 In the experiments of CIFAR datasets, LECE performs too badly without using TS.**
>
> From the experiments it can be concluded that on datasets with few samples per class, LECE alone is not strong enough to compete with methods with parametric assumptions.
> However, LECE has its merits being less constrained than its competitors, and therefore it can offer improvements on top of parametric methods, wherever they are limited by their parametric family.
> Additionally, the violations of the LECE assumption are likely to diminish when the calibration errors become smaller, e.g. on top of temperature scaling.
> We have added the above explanation to the paragraph "Composition with parametric methods" in Section 3.2.
>
> **(2.2 continues) And it is hard to attribute the nice performance of TS+LECE to LECE. Since in certain cases, TS could achieve fairly well. Adding LECE has no further improvements.**
>
> In nearly all cases (except confidence ECE on DenseNet40 C-100), applying LECE on top of TS offers improvements.
> These improvements might seem minor when evaluated with log-loss (Table 5), but this is because log-loss includes both refinement loss and calibration loss. When evaluating with confidence ECE or classwise ECE (Tables 3 and 4), then the relative differences are much bigger, and TS+LECE does offer considerable improvements over TS.
>
> **2.3 More detailed explanations on why adding LECE requires TS might be needed, for the experiments on CIFAR datasets.**
>
> This we addressed above when answering question 2.2.

---

> ### Author Response · Authors · 2023-04-12
> **Response to Reviewer QVr6 (part 2)**
>
> **2.4 When looking into the ablation study (Table 7), please correct me if my understanding is incorrect: personally, it seems that the reason why the authors report the performance of TS+LECE rather than others, i.e., $LECE_\{euc\}$ or $TS+LECE_\{euc\}$ is because of the performance on the dataset for evaluation? For me, I feel like this is kind of cherry-picking. In practice, why do the authors decide on reporting $TS+LECE$ instead of $TS+LECE_\{euc\}$ for comparing with other methods?**
>
> Cross-validation performed on the training data showed that $TS+LECE$ (with KL-divergence) was better than other considered methods ($LECE, TS+LECE, TS+LECE_\{euc\}, LECD, TS+LECD$).
> Therefore, TS+LECE could be concluded to be the best-performing method without looking at the test set results, thus avoiding cherry-picking.
> The ablation study was performed to confirm this result on the test data.
>
> **Suggestion 1: "Probabilistic classifiers take as input some data and produce as output probability distributions over classes." --> It might be better to say "Probabilistic classifiers take some data as input and produce probability distributions over classes as output."**
>
> Corrected following the suggestion.
>
> **Suggestion 2: "It is possible that a probabilistic classifier is not well-calibrated and produces distorted probabilities. A classifier is considered to be calibrated if its predicted probabilities are in correspondence with the true class distribution." --> It might be better to switch the order of these two sentences.**
>
> Corrected following the suggestion.
>
> **Suggestion 3: A toy example for the illustration of three definitions in Section 2.2 could be much better.**
>
> We have added a toy example.
>
> **Suggestion 4: In Section 2.4, more introduction about proper scoring rules might be better.**
>
> We have now added more explanations about proper scoring rules to the end of Section 2.4 and at the end of the first paragraph of Section 2.5.
>
> **Notation 1 (Expectation): on page 6, when talking about expectations, it could be better to what the expectation is with respect to, making the presentation more clear, i.e., $\mathbb{E}_\{?\}[\hat{CE}_\{neigh\}(\hat{p})$.**
>
> We have now explicitly specified what all the expectations are taken over.
>
> **Suggestion 5: a brief introduction of the synthetic dataset at the end of page 6 could be much better. Or put the visualization of the assumptions in the experiment section?**
>
> Thank you for this great suggestion. We added a brief introduction of the synthetic data set when first referencing Figure 1.
> We also split the figure into two: now Figure 1 only contains the subfigures referenced in paragraph "Visualisation of the assumptions"; the other subfigures were moved to Figure 2 in the experiments section.
> Note that the subfigure with the true calibration map is included for convenience in both Figure 1 and Figure 2.
>
> **Question 1: is the visualization (calibration map) a novel visualization tool for comparisons between post-hoc calibrations?**
>
> No, it has been used previously, for example, in the article about intra-order preserving functions (Rahimi et al., 2020).
>
> **Question 2: Figure 1 is kind of hard for me to follow. The differences among certain figures (except for (a)) are not quite straightforward to catch. Can authors adopt two sub-figures and explain their differences in more detail (i.e., (d), (e), (f))? Besides, more explanations on settings of the sub-figure are appreciated, i.e., what information is the color conveying. As for the differences between calibration maps, can they be quantified in numeric values?**
>
> See the above answer to Suggestion 5.
> We added some additional sentences pointing to some details of the subfigures of Dirichlet, LECE, and the true calibration map.
> The main difference between the subfigures are the calibration arrows near small values of $\hat{p}_1$, where Dirichlet calibration fails to imitate the calibration arrows of the true calibration map, but LECE manages OK.
>
> The meaning of colors is explained in the paragraph "Visualisation of the assumptions" under Section 3.1.
>
> Numeric quantification of the differences between calibration maps is provided in Table 1.
>
> **Typo 1: on page 8, "The LECE assumption outperformed the LECD assumption on the real experiments also". "Also" $\rightarrow$ well.**
>
> Fixed.
>
> **Suggestion 6: the first sentence of section 3.2 reads weird.**
>
> Removed this sentence.
>
> **Notation 2 ($\hat{p}_\{1i\}$): some notations are not well defined/introduced, i.e., in the introduction of KL-divergence on page 9, $\hat{p}_\{1i\}$ is not well-defined.**
>
> In this context, we have changed $\hat{\mathbf{p}}_1$ and $\hat{\mathbf{p}}_2$ into $\hat{\mathbf{p}}$ and $\hat{\mathbf{p}}'$, respectively.
> Due to this, $\hat{p}_\{1i\}$ is now replaced by $\hat{p}_\{i\}$ and $\hat{p}_\{2i\}$ by $\hat{p}'_\{i\}$.

---

> ### Author Response · Authors · 2023-04-12
> **Response to Reviewer QVr6 (part 3)**
>
> **Notation 3 ($\hat{c}(\hat{p})$): at the end of paragraph "Thresholding tiny probabilities", the notation is not consistent, i.e., $\mathbf{\hat{c}}(\mathbf{\hat{p}})$ and $\hat{c}(\mathbf{\hat{p}})$
>  and
> .**
>
> We have changed the location of the subindex $i$ from $\hat{c}(\mathbf{\hat{p}})_i$ into $\hat{c}_i(\mathbf{\hat{p}})$ and added the definition $\mathbf{\hat{c}}(\cdot)=(\hat{c}_1(\cdot),\dots,\hat{c}_m(\cdot))$.
>
> **Typo 2: in the last sentence of the paragraph "Composition with parametric methods", white space is needed in the front.**
>
> Fixed.
>
> **Exp 1: can authors try 2 other true calibration functions to see if the performance of LECE is still consistently better than DIR, in the synthetic experiments?**
>
> Addressed above, when answering question 2.1.
>
> **Exp 2: why do authors choose $k=500,t=0$ in the synthetic dataset, and what are the candidate values of $k,t$ in this synthetic experiment?**
>
> Addressed above, when answering question 2.1.
>
> **Exp 3: at the end of section 4.1.3, can the authors explain how the conclusion "the proposed LECE calibration method depicted in Figure 1e learns the most similar transformation to the true calibration map" is made? Is it numerically tested?**
>
> Numeric quantification of the differences between calibration maps is provided in Table 1.
>
> **Exp 4: in experiments of CIFAR datasets, for TS+LECE, what temperature scaling parameter is adopted? And how is it selected?**
>
> Temperature scaling in TS+LECE is trained the same way as in just TS: the temperature parameter is optimized on the logits of the uncalibrated classifier with log-loss and BFGS optimization algorithm.
> The selected temperature scaling parameters are the same and are the following:
> CIFAR-10 dense: 2.88, wide32: 3.10, resnet: 2.40; CIFAR-100 dense: 3.19, wide32: 3.04, resnet: 2.31.
>
> **Exp 5: for the comparison between TS, and TS+LECE, is TS+LECE always better than TS?**
>
> Yes, in all cases except confidence ECE on DenseNet-40 CIFAR-100 and classwise ECE on ResNet-110 CIFAR-10.
>
> **Exp 6: in experiments of CIFAR datasets, while searching for optimal parameters, do the authors have any observations on the pattern of good parameters?**
>
> Good values seem to be between $0.01,\dots,0.06$ for $q$ (translating to the number of neighbors $k$ between
> $0.01\cdot 5000=50,\dots,0.06\cdot 5000=300$).
> For the threshold $t$, for just LECE $t=0$ is always best both on CIFAR-10 and CIFAR-100, for TS+LECE the optimal values are between $0.0025,\dots, 0.02$.
> The optimal values are provided in Tables 10 and 11.
>
> **Exp 7: More detailed explanations on why adding LECE requires TS might be needed, for the experiments on CIFAR datasets.**
>
> This we addressed above when answering question 2.2.
>
> **Exp 8: experiment details on how the statistics are calculated (i.e., the evaluation dataset, etc), are needed, for CIFAR experiments.**
>
> The precomputed logits of the CIFAR experiments were provided by Kull et al. (2019), and exactly the same validation and test set split of the logits was used as in the experiments of Kull et al. (2019) and Rahimi et al. (2020).
> We improved our wording regarding this detail in Section 4.2.1.
>
> **Exp 9: ...it seems that the reason why the authors report the performance of TS+LECE rather than others, i.e., $LECE_\{euc\}$
>  or $TS+LECE_\{euc\}$ is because of the performance on the dataset for evaluation? For me, I feel like this is kind of cherry-picking. In practice, why do the authors decide on reporting
>  $TS+LECE$ instead of $TS+LECE_\{euc\}$ for comparing with other methods?**
>
> Addressed above when answering question 2.4.

---

> > ### Comment · Reviewer_QVr6 · 2023-04-17
> > **Post-Rebuttal**
> >
> > Thanks authors for addressing my concerns and taking my suggestions into account.
> >
> > Most of my concerns have been addressed in the rebuttal. Besides, I think the explanations regarding the major weakness (performances on CIFAR datasets) make sense to me. I don't have additional concerns for now. Thanks again for all your time on the detailed responses.

---

### Review · Reviewer_Fi18 · 2023-03-29

**Summary Of Contributions:**

This paper uses the assumption that nearby points on the probability simplex have similar calibration mappings to propose a post-hoc calibration method based on the nearest neighbor algorithm. They present two assumptions on the local neighborhood of a point on a probability simplex: 1) LECE assumes nearby points have equal calibration errors, and 2)  LECD assumes nearby points have the same class distribution. The authors show that their LECE assumption leads to an unbiased estimation of calibration error. Then the authors discuss the Relation of their work to previous works and provide some practical considerations to enhance the results. The results show that the proposed method is comparable to other state-of-the-art methods.

**Audience:**

Yes

**Broader Impact Concerns:**

I see no concern about broader impact of this work.

**Claims And Evidence:**

Yes

**Requested Changes:**

Please see the Strengths And Weaknesses section.

**Strengths And Weaknesses:**

**Pros**
---------
+ The paper is easy to read.

+ The proposed method is relatively simple, technically sound, and has strong results.

+ Relation to previous works and discussing the drawbacks and fair comparison to previous methods are valuable.

**Cons**
---------
- The motivation for the proposed method needs to be improved in the abstract.

- The proposed method has the main drawbacks of non-parametric: mainly sensitivity to noise and being computationally expensive. These may also need to be discussed in the main text.
- The accuracies of the models after calibration are not present in the experiments.

- Some drawbacks limit the applicability of the proposed method. As the number of classes increases, the method becomes computationally expensive and less reliable due to high dimensionality. Similarly, the number of validation samples, whether small or large, affects the results. Although these issues are discussed to some extent in the paper, conducting experiments by varying the number of validation samples or using a greater number of classes (such as ImageNet) and reporting the computation times would further strengthen the paper.

---

> ### Author Response · Authors · 2023-04-12
> **Response to Reviewer Fi18**
>
> Thank you for your comments! We have taken your feedback into account and uploaded a new version of the paper with changes highlighted in blue. Below are our responses to your comments.
>
> **The motivation for the proposed method needs to be improved in the abstract.**
>
> Thank you, we have now improved the abstract.
>
> **The proposed method has the main drawbacks of non-parametric: mainly sensitivity to noise and being computationally expensive. These may also need to be discussed in the main text.**
>
> Indeed, non-parametric methods including LECE are sensitive to noise due to their non-parametric assumptions about the data.
> In LECE, hyperparameter tuning addresses the sensitivity by choosing the number of neighbours, hitting the best tradeoff of bias and variance, as briefly discussed at the end of Section 3.1.
> In particular, if the noise level is too high, then LECE chooses the threshold t=1.0, which effectively sets the calibration map to be the identity function.
>
> Computational and memory complexity discussion has been added to Section 3.2.
> Discussion about the running times of LECE has been added to Section 4.2.5.
>
> **The accuracies of the models after calibration are not present in the experiments.**
>
> Added Table 6 presenting the accuracies of the models and a small corresponding discussion.
>
> **...conducting experiments by varying the number of validation samples or using a greater number of classes (such as ImageNet) and reporting the computation times would further strengthen the paper.**
>
> Given the limited time for responding to the reviews, we have currently decided not to extend the experiments to ImageNet.
> While running our method on ImageNet is relatively straightforward, running comparisons with all of the other methods would require adapting the source code of some of these methods and therefore requires a substantial investment of time.

---

> > ### Comment · Reviewer_Fi18 · 2023-04-29
> > **Post-rebuttal response to authors**
> >
> > I would like to thank the authors for addressing most of my concerns. I still think that an experiment with larger number of classes makes the paper more valuable in the final revision.
> >
> > **Minor issue** one the drawbacks of 1-vs-all methods for posthoc calibration is mentioned to be the normalization step. However, the normalization after thresholding in LECE is also having the same issue.

---

### Review · Reviewer_xZn1 · 2023-04-11

**Summary Of Contributions:**

The LECD principle is inherent in many post-hoc calibration methods such as temperature scaling, histogram binning, isotonic regression, fixed-width binning, Dirichlet calibration, etc. The LECD principle being: locally (with respect to the pre-hoc classifier), the true label distribution is approximately equal.

The authors propose an alternative LECE principle: locally (with respect to the pre-hoc classifier), calibration errors are approximately equal. New calibration methods can be derived based on the LECE principle, and these seem to outperform existing methods that are based on the LECD principle. This is demonstrated through an illustration on synthetic data, and standard baselining on CIFAR-10 and CIFAR-100 (with deepnets).

**Audience:**

Yes

**Claims And Evidence:**

Yes

**Requested Changes:**

### Major:

- Please add one or two 1-v-rest baselines in the empirical comparison (perhaps based on histogram binning/Platt scaling/isotonic regression/ Bayesian binning in quantiles, etc).
- Histogram binning has bins with equal mass, whereas the method you have used in experiments is equal-width binning. Please refer to the method you are using as equal-width binning to avoid confusion.
- Supplement all tables with meaningful captions following standard scientific conventions. In general, the experiments section is currently not well-organized: float placement is not standard (middle of pages instead of top), and the lists of methods split across pages too many times. Not everything needs to be standard but too many deviations are not ideal.
- For instance, Table 1 appears suddenly in the middle of Sec 4.1 without a caption and without introducing the acronymized methods. The table should perhaps be on top of the following page, and a caption should be included summarizing the takeaway message, describing what the subindices are, what "true" means in the last column, etc (I mention the things that were unclear to me, but other things could be unclear to other readers, so please include a comprehensive caption).
- Sec 4.1: you mention averaging over 10 seeds, but no standard deviation is reported. Thus, I am not sure of the significance of the results in Table 1. Since it is a synthetic experiment, I suggest derandomizing with 1000 simulations and reporting the (very small) standard deviation to make the results almost 100% conclusive.
- What splits were used for the CIFAR post-hoc experiments? If splits were standard, refer to where these were obtained from. If splits were random, report standard deviations in the results.

### Minor but strongly recommended

- The fixed-width binning in 3-dimensions that you used was initially suggested in the paper by Vaicenavicius et al. (2019), so do cite this contribution.
- The title is not very informative since "nonparametric multiclass calibration" is quite broad and the qualification "simple" is subjective. Please provide a more informative title without the subjective qualifier.
- Page 5: the description of intra order-preserving functions can be improved.
- The motivation/background for "thresholding tiny probabilities" could be made clearer. The description makes a number of statements without theoretical/empirical justification or referencing other work: "these build-in errors become very large proportional to the probability", "LECE method produces output smaller than 0". The method of correction is also not well-unmotivated; why retain the original probability and not just set c-hat(p-hat) = 0? (In contrast, the paragraph immediately after, "Composition with parametric methods" is very clear! Citations and independent justifications are provided, and an experiments section is also referenced.)

### Minor and up to the authors

- I believe when you say 'locally' in the paper, you mean with respect to the pre-hoc classifier. This may be useful to qualify explicitly.
- If the authors can show on an illustrative synthetic example that LECE provably does better than LECD, that would be a very useful addition to the paper.

**Strengths And Weaknesses:**

### Strength:

I liked the proposed LECE principle. It is simple, makes complete sense, and as far as I know has not been studied before. Figure 1 is quite useful and the paper is easy to read until the end of Section 3. Experiments suggest the method works well. The authors go beyond only comparing to other calibration methods, also studying the following aspects:

- the effect of scaling before applying LECE,
- two notions of distance and their parameters,
- dealing with near-zero probabilities by thresholding,
- behavior with respect to multiple calibration notions as well as proper scoring rules.

Once revised and published, I think the LECE method/principle will be a valuable addition to the calibration community.

### Weaknesses:

- No 1-v-rest baselines are reported in the empirical comparisons. While some reasons are briefly indicated on page 4 for ignoring 1-v-rest methods ("There are two ... deforms the calibrated probabilities"), no citations or experiments are included. I am aware of at least a couple of papers (https://arxiv.org/pdf/2107.08353.pdf, https://arxiv.org/pdf/2006.13092.pdf) where it was shown that 1-v-rest methods can outperform some full-vector scaling methods that you considered.

- Some presentation aspects need to be strengthened, as elaborated in the "requested changes" segment. Particularly, the experiments and tables can be improved significantly. Other issues were the misnomer of "histogram binning" for fixed-width binning, lack of good motivation for thresholding small probabilities, and not reporting standard deviations (or other measures of statistical uncertainty) in the empirical findings.

- The technical discussion around LECE on page 6 is limited and informal. I think it is better to be informal and brief than to have meaningless theory, so I have not requested a revision on this front.

---

> ### Author Response · Authors · 2023-04-18
> **Response to Reviewer xZn1 (part 1)**
>
> Thank you for your review and constructive feedback addressing the paper's weaknesses.
>
> Note that some of the requested changes were already implemented in the article revision posted on April 13th in response to reviewers Fi18, QVr6, and eGb8.
> Changes done in the revision on April 13th are marked in blue, while changes in response to your review are marked in brown.
>
> ### Weaknesses
>
> **No 1-v-rest baselines are reported in the empirical comparisons. While some reasons are briefly indicated on page 4 for ignoring 1-v-rest methods ("There are two ... deforms the calibrated probabilities"), no citations or experiments are included...**
>
> Thank you for pointing this out. We have now added two 1-v-rest baselines to the real experiments and expanded on the background of 1-v-rest methods in Section 2.5, adding two citations.
> See our more detailed response in the requested changes section below.
>
> **Some presentation aspects need to be strengthened, as elaborated in the "requested changes" segment...**
>
> Thank you for highlighting these weaknesses!
> See our response to all these details in the requested changes section below.
>
> **The technical discussion around LECE on page 6 is limited and informal. I think it is better to be informal and brief than to have meaningless theory, so I have not requested a revision on this front.**
>
> We reformulated the LECD and LECE assumptions more formally in Section 3.1, as this weakness was also highlighted by reviewer QVr6.
>
> ### Requested Changes. Major
>
> **Please add one or two 1-v-rest baselines in the empirical comparison (perhaps based on histogram binning/Platt scaling/isotonic regression/ Bayesian binning in quantiles, etc).**
>
> We have now added 1-v-rest isotonic regression (IR) and the same method applied on top of temperature scaling (TS+IR) as baselines to our experiments.
> We also tested a few more 1-v-rest methods: histogram binning, histogram binning with the LECE assumption instead of the LECD assumption, and both histogram binning methods applied in composition with TS.
> For all histogram binning methods, we applied them with equal-sized (i.e. equal-mass) bins where the number of bins was found with cross-validation.
> None of the 1-v-rest methods outperformed TS+LECE on log-loss, confidence ECE or classwise ECE.
> To keep the tables readable, we added just IR and TS+IR to the article, as isotonic calibration is a simple non-parametric method which we did not yet have in the comparisons.
>
> **Histogram binning has bins with equal mass, whereas the method you have used in experiments is equal-width binning. Please refer to the method you are using as equal-width binning to avoid confusion.**
>
> Thank you for bringing up this detail!
> We have made several small changes to increase clarity.
> * When first mentioning the binning schemes in Section 2.4, we now mention what we mean by equal-sized bins (aka equal-mass bins) and equal-width bins.
> * When first introducing the histogram binning method in Section 2.5, we now mention that it is meant to be applied with equal-sized bins.
> * When first introducing Figure 1 in Section 3.1, we now mention that for Figure 1, we have applied histogram binning with equal-width bins, not with equal-sized bins as done in the original article. The reason was to have a figure with equal triangles instead of triangles of many different sizes.
> * When referencing the synthetic experiment (or Figure 1), we now always refer to the method as "histogram binning with equal-width bins": it was also done previously in most places, but not in the caption of Figure 1.

---

> > ### Author Response · Authors · 2023-04-18
> > **Response to Reviewer xZn1 (part 2)**
> >
> > **Supplement all tables with meaningful captions following standard scientific conventions. In general, the experiments section is currently not well-organized: float placement is not standard (middle of pages instead of top), and the lists of methods split across pages too many times. Not everything needs to be standard but too many deviations are not ideal.
> > For instance, Table 1 appears suddenly in the middle of Sec 4.1 without a caption and without introducing the acronymized methods.
> > The table should perhaps be on top of the following page, and a caption should be included summarizing the takeaway message, describing what the subindices are, what "true" means in the last column, etc (I mention the things that were unclear to me, but other things could be unclear to other readers, so please include a comprehensive caption).**
> >
> > We have elaborated the captions of all tables:
> > * added a key takeaway message to all tables,
> > * reference where to find details about the acronymized methods,
> > * mention the meaning of subindices and bold text,
> > * mention any additional table-specific details (e.g. the meaning of "true" in Table 1).
> > In addition, we have improved the placement of tables and the bullet-lists which previously were cut in too many places.
> >
> > **Sec 4.1: you mention averaging over 10 seeds, but no standard deviation is reported. Thus, I am not sure of the significance of the results in Table 1. Since it is a synthetic experiment, I suggest derandomizing with 1000 simulations and reporting the (very small) standard deviation to make the results almost 100\% conclusive.**
> >
> > We have now increased the test set size from 10000 to 100000 and the number of simulations from 10 to 100, and have included the standard deviations in the table.
> >
> > **What splits were used for the CIFAR post-hoc experiments? If splits were standard, refer to where these were obtained from. If splits were random, report standard deviations in the results.**
> >
> > The precomputed logits of the real experiments were provided by Kull et al. (2019), and the same validation and test set split of the logits was used as in the experiments of Kull et al. (2019) and Rahimi et al. (2020).
> > We have added the previous sentence to Section 4.2.1.
> >
> > ### Requested Changes. Minor but strongly recommended
> >
> > **The fixed-width binning in 3-dimensions that you used was initially suggested in the paper by Vaicenavicius et al. (2019), so do cite this contribution.**
> >
> > Thanks for the detail! We have added the citation in Section 3.1.
> >
> > Also, as a note, to avoid confusion for any readers, Vaicenavicius et al. (2019) used the fixed-width binning in 3-dimensions for a reliability diagram, but in our article in Figure 1a, a calibration method is depicted, not a reliability diagram.
> > Even though the two are technically the same, in one case, the result is used for calibration evaluation, and in the other case, for representing the calibration map.
> >
> > **The title is not very informative since "nonparametric multiclass calibration" is quite broad and the qualification "simple" is subjective. Please provide a more informative title without the subjective qualifier.**
> >
> > We have now renamed the paper; the new title is ``Assuming locally equal calibration errors for non-parametric multiclass calibration''.
> >
> > **Page 5: the description of intra order-preserving functions can be improved.**
> >
> > We have expanded the description of intra order-preserving functions.

---

> ### Author Response · Authors · 2023-04-18
> **Response to Reviewer xZn1 (part 3)**
>
> **The motivation/background for "thresholding tiny probabilities" could be made clearer. The description makes a number of statements without theoretical/empirical justification or referencing other work: "these build-in errors become very large proportional to the probability", "LECE method produces output smaller than 0". The method of correction is also not well-unmotivated; why retain the original probability and not just set c-hat(p-hat) = 0? ...**
>
> We have improved the paragraph about thresholding in Section 3.2 to include justifications, an example, and a reference to the threshold values used in the real experiments.
>
> The updated paragraph is as follows
>
> > **Thresholding tiny probabilities** A problem inherent to the non-parametric LECE (and LECD) calibration method is its inability to work well for tiny probabilities.
> This is because the method uses an estimator, which has some built in errors coming from its bias and/or variance.
> For class probabilities that are very near to 0, these errors of the estimator become very large proportionally to the probability.
> To see this, consider a true class probability $p_i$ estimated based on $k$ neighbours. In the ideal case where all the neighbours have the same true label distribution, the variance of this estimator is $\frac{p_i(1-p_i)}{k}$. Hence the estimator's relative error (standard deviation divided by the true value) is $\frac{\sqrt{p_i(1-p_i)}}{\sqrt{k}}/p_i=\frac{\sqrt{1-p_i}}{\sqrt{p_i\cdot k}}$ which becomes increasingly large when $p_i$ gets small.
> This could even lead to situations, where the LECE method produces output that is smaller than 0 for some classes and could no longer be interpreted as probabilities.
> For example, consider $\hat{p}_i=0.01$ and suppose the average prediction of its $k$ neighbors is $\bar{p}=0.03$ and the average label $\bar{y}=0.01$.
> In that case, the calibration error estimate is $\widehat{CE}_\{neigh\}(\hat{p_i})=0.03-0.01=0.02$ and the calibrating transformation would be $\hat{c}(\hat{p}_i)=\hat{p}_i - \widehat{CE}_\{neigh\}(\hat{p_i}) = 0.01 - 0.02=-0.01$, which is no longer on the probability simplex.
> Therefore, to overcome this problem with small probabilities, we opted to introduce one more parameter to the calibration method: a threshold value $t\in\mathbb{R}$ which sets a lower limit when to apply the method.
> For any class probability smaller than $t$, we do not apply the method.
> As the true class probability $p_i$ is unknown, then instead we apply this threshold on both the uncalibrated prediction $\hat{p}_i$ and the corresponding would-be-calibrated prediction $\hat{c}(\hat{p}_i)$.
> More precisely, given the prediction vector $\mathbf{\hat{p}}$, and the would-be-calibrated prediction vector $\mathbf{\hat{c}}(\mathbf{\hat{p}})=\mathbf{\hat{p}}-\widehat{CE}_\{neigh\}(\mathbf{\hat{p}})$, if for the $i$-th class $\hat{p}_i\leq t$ or $\hat{c}_i(\mathbf{\hat{p}})\leq t$, then we set $\hat{c}_i(\mathbf{\hat{p}})=\hat{p}_i$, where $\mathbf{\hat{c}}(\cdot)=(\hat{c}_1(\cdot),\dots,\hat{c}_m(\cdot))$.
> Thresholding can cause the final output to no longer sum to 1, so to solve this, as a final step we divide the output vector by its sum.
> As shown by optimal threshold values chosen by hyperparameter tuning in the real experiments in Table 11, the LECE method chooses small threshold values ranging from $t=0$ to $t=0.02$.
>
> ### Requested Changes. Minor and up to the authors
>
> **I believe when you say 'locally' in the paper, you mean with respect to the pre-hoc classifier. This may be useful to qualify explicitly.**
>
> That is a good point! We have now added the following sentence when introducing the LECE assumption:
> ``Note that the term ‘locally‘ is often used to refer to neighborhoods in the original feature space, whereas we
> consider neighborhoods in the simplex.''
>
> **If the authors can show on an illustrative synthetic example that LECE provably does better than LECD, that would be a very useful addition to the paper.**
>
> The current illustrative experiment on synthetic data is meant to fulfill this purpose.
> The histogram binning with equal-width bins with the LECE assumption outperforms the classical version with the LECD assumption (Figure 1 and Table 1).
>
> However, this should not be taken as proof that the LECE assumption is always more reasonable than the LECD assumption. The success of a calibration method making one or the other assumption depends heavily on the calibration task as well as on the calibration method itself.

---

### Author Response · Authors · 2023-04-12
**Summary of revised manuscript**

We are very thankful for the constructive and helpful feedback from all 4 reviewers.
We have now replied to reviewers Fi18, QVr6, and eGb8, and modified the paper accordingly (new and modified text marked in blue). We will respond to the reviewer xZn1 within 1 week, because we only received the review 1 day before our responding deadline.

We made changes to nearly all sections, but the most significant changes are the following:

* The abstract was improved (in response to reviewers eGb8, Fi18).

* A toy example about the different definitions of calibration was added to Section 2.2 (reviewer QVr6).

* More introduction about proper scoring rules added to Sections 2.4 and 2.5 (reviewer QVr6).

* Reformulated the LECD and LECE assumptions more formally in Section 3.1 (reviewer QVr6).

* Split and rearranged Figure 1 into Figures 1 and 2 to match the logic of the narrative (reviewer QVr6).

* Added details about the computational and memory complexity (Section 3.2) as well as empirical running times (Section 4.2.5) (reviewers eGb8, Fi18).

* Added pseudo-code of the LECE calibration method as Algorithm 1 at the end of Section 3 (reviewer eGb8).

* Added standard deviations to Table 1 (reviewer eGb8).

* Added Table 6 reporting accuracy of the methods (reviewer Fi18).

* Made many smaller edits following the feedback from the reviewers.

We are happy to respond to any further comments or questions.

---

### Decision · Action_Editors · 2023-05-11

**Recommendation:** Accept as is

**Comment:**

The paper studies the problem of calibrating multi-class classifiers, a fundamental problem of broad interest to the community. It explicates conditions underpinning the success of calibration that have hitherto not been discussed directly. These are used to devise a new calibration scheme with strong empirical performance. Reviewers unanimously agreed that the paper makes an interesting contribution that would be valuable for the TMLR audience. From our reading, we concur with this view. We are thus pleased to recommend the paper for acceptance.

For the final version, we do agree with one reviewer's suggestion that the authors consider reducing the length of the the main body.

**Audience:**

All reviewers agreed the paper would be of interest to the TMLR community.

**Claims And Evidence:**

All reviewers agreed the paper makes clear claims that are backed by convincing evidence, both analytical and empirical.